



# Brown carbon's emission factors and optical characteristics in household biomass burning: Developing a novel algorithm for estimating the contribution of brown carbon

Jianzhong Sun[1,2], Guorui Zhi[1,*], Regina. Hitzenberger[3], Yingjun Chen[4], Chongguo Tian[5],

[1]State Key Laboratory of Environmental Criteria and Risk Assessment, Chinese Research Academy of
 Environmental Sciences, Beijing 100012, China
[2]Shangrao Normal University, Shangrao 334001, China
[3]University of Vienna, Faculty of Physics, Boltzmanngasse 5, 1090 Vienna, Austria
[4]Shanghai Key Laboratory of Atmospheric Particle Pollution and Prevention (LAP3), Department of
 Environmental Science & Engineering, Institute of Atmospheric Sciences, Fudan University,
 Shanghai 200433, China
[5]Key Laboratory of Coastal Environmental Processes and Ecological Remediation, Yantai Institute of
 Coastal Zone Research, Chinese Academy of Sciences, Yantai 264003, China

*Correspondence to*: Guorui Zhi (zhigr@craes.org.cn)





**Abstract**. Recent studies have highlighted the importance of brown carbon (BrC) in various fields,
particularly relating to climate change. The incomplete combustion of biomass in open and contained
burning conditions is believed to be a significant contributor to primary BrC emissions. So far, few
studies have reported the emission factors of BrC from biomass burning, and few studies have
specifically addressed which form of light absorbing carbon, such as black carbon (BC) or BrC, plays a
leading role in the total solar light absorption of biomass burning. In this study, the optical integrating
sphere (IS) approach was used, with carbon black and humic acid sodium salt as reference materials for
BC and BrC, respectively, to distinguish BrC from BC on the filter samples. Eleven widely used
biomass types in China were burned in a typical stove to simulate the real household combustion
process. (i) Large differences existed in the emission factors of BrC ($EF_{BrC}$) among the tested biomass
fuels, with a geomean $EF_{BrC}$ of 0.71 g/kg (0.24, 2.18). Both the plant type (herbaceous or ligneous) and
burning style (raw or briquetted biomass) might influence the value of $EF_{BrC}$. (ii) The calculated annual
BrC emissions from China's household biomass burning amounted to 712 Gg, higher than the
contribution from China's household coal combustion (592 Gg). (iii) The average absorption Ångström
exponent (AAE) was (2.46 ± 0.53), much higher than that of coal-chunks combustion smoke (AAE =
1.30 ± 0.32). (iv) For biomass smoke, the contribution of absorption by BrC to the total absorption by
BC + BrC across the strongest solar spectral range of 350–850 nm ($F_{BrC}$) was 50.8%. This was nearly
twice that for BrC in smoke from household coal combustion (26.5%). (v) Based on this study, a novel
algorithm was developed for estimating the $F_{BrC}$ for any combustion sources ($F_{BrC}$ = 0.5519lnAAE +
0.0067, $R^2$ = 0.999); the $F_{BrC}$ value for global entire biomass burning (open + contained) ($F_{BrC-entire}$) was
64.5% (58.5–69.9%). This corroborates the dominant role of BrC in total biomass burning absorption.
Therefore, BrC is not optional but indispensable when considering the climate energy budget,
particularly for biomass burning emissions (contained and open).



## 1 Introduction

Brown carbon (BrC) refers to the fraction of organic carbon (OC) that is light-absorbing, with a
pronounced wavelength dependence of absorption (Kirchstetter et al., 2004; Bosch et al., 2014;
Chakrabarty et al., 2014; Mo et al., 2017; Jiang et al., 2018; Sun et al., 2018). Recent studies have
highlighted the importance of BrC in not only atmospheric chemistry, air quality and human health, but
also for climate change (Chakrabarty et al., 2010; Huang et al., 2018; Yan et al., 2018; Han et al., 2020).
The light absorption of BrC is more emphasised towards short wavelengths, particularly in the
ultraviolet (UV) range, on account of there being a larger spectral dependence for BrC than for BC
(IPCC, 2014; Pokhrel et al., 2017; Li et al., 2018; Ferrero et al., 2020). By calculating the radiative
forcing (RF) of BrC at the surface and at the top of the atmosphere, Park et al. (2010) found that more
than 15% of the total RF caused by light absorbing carbon (LAC, including BrC and BC) could be
attributed to BrC. Yao et al. (2017) demonstrated that a positive direct radiative effect (DRE) of
absorption ($+0.21$ W·m$^{-2}$) was caused by BrC-containing organic aerosols from the burning of crop
residues in East China during the summer harvest season. This is indicative of the negative effects on
not only air quality, but also on climate. Pokhrel et al. (2017) found that the absorption of BrC at
shorter visible wavelengths was equal to or greater than that of BC.
The incomplete smouldering combustion of biomass in open environments or contained stoves is
a major contributor to primary BrC emissions (Lukács et al., 2007; Chakrabarty et al., 2010; Hecobian
et al., 2010; Chakrabarty et al., 2013). High gas and particle emissions have often been observed during
these combustion processes (Kirchstetter et al., 2004; Chen and Bond, 2010; Bosch et al., 2014;
Budisulistiorini et al., 2017). Ground-based observations and model simulations have revealed that in
some regions with high biomass consumption intensities, such as South America, South Asia, Africa,
Russia, China, and India, high levels of BrC (10–35 mg·m$^{-2}$) are found in the atmosphere (Arola et al.,
2011; Feng et al., 2013; Huang et al., 2018). In these regions, the climatic effects of BrC are expected
to be stronger than in other regions.
In China, biomass burning contributes a substantial quantity of carbonaceous particles, along with
many other air pollutants. The available emission inventories show that approximately 20% of primary
fine particulate matter (PM$_{2.5}$) originates from biomass burning (open and contained) (Yao, 2016). Zong



et al. (2017) used the Positive Matrix Factorisation (PMF) method, linked with radiocarbon analysis, to
conduct a source apportionment study of $PM_{2.5}$ at a regional background site in northern China. They
identified that biomass combustion comprised a significant contribution (19.3%) of atmospheric $PM_{2.5}$.
Cheng et al. (2013) confirmed the significance of biomass burning in air pollution, finding that
approximately 50% of OC and elemental carbon (EC) in Beijing were associated with biomass burning
processes. It is also suggested that more biomass is burned in stoves than in open fields, due to China's
continued efforts to prevent and control forest fires and the burning of field stalks (Tian et al., 2011;
Zhi et al., 2015a; Cheng et al., 2016). Hence, more attention should be paid to the household sector
than to open burning, as far as biomass-related emissions are concerned in China. In addition, unlike
other regions where firewood often plays a major role as a biomass fuel, China has more access to
agricultural waste (e.g. maize straw, wheat straw, and rice straw) for household heating/cooking
purposes (Huang et al., 2012; Shen et al., 2013; Chen et al., 2015a). This suggests that studies of BrC
originating from China's household biomass fuel combustion should consider as many biomass fuel
varieties as possible, so that the actual characteristics of BrC emissions can be comprehensively
investigated and represented.
The available literature dealing with BrC from biomass burning in China to date has generally
focussed on ambient observation (Arola et al., 2011; Chakrabarty et al., 2014; He et al., 2017; Zhao et
al., 2018) and modelling (Gustafsson et al., 2009; Feng et al., 2013) of the basic characteristics of
atmosphere, such as the concentrations and temporal and spatial distributions. Few studies have
addressed the typical sources of emission characteristics (Lin et al., 2017; Mo et al., 2017; Phillips et al.,
2017; Rawad et al., 2018; Sumlin et al., 2018; Xu et al., 2018; Zhang et al., 2018). Even though a few
studies have collected emission samples at some sources, the objectives of these studies was to further
understand the general properties of water soluble organic carbon (WSOC) or methanol soluble organic
carbon (MSOC) (Cheng et al., 2013, 2016; Lin et al., 2017; Phillips et al., 2017; Wu et al., 2019; Yan et
al., 2020). Consequently, there is a lack of knowledge regarding source emission strengths (emission
factors; EFs) and regarding how BrC's role of absorption differs relative to BC (Lack et al., 2012;
Healy et al., 2015; Washenfelder et al., 2015; Srinivas, et al., 2016; Zhang et al., 2016). An intensive
study on BrC from China's household biomass emission sources is therefore necessary to provide





insight into both the EFs and light absorption properties of particulate emissions.

In the present study, eleven biomass fuels that are widely used in China were burned in an

ordinary stove, to simulate domestic burning practices. Particulate emissions were collected by quartz
filters to measure the EFs of BrC ($EF_{BrC}$) and BC ($EF_{BC}$) for China's household biomass burning, for
investigating the spectral characteristics of absorption by BrC and estimating the contribution of BrC to
total light absorption by BC + BrC across a broad solar spectral range (350–850 nm). The integrating
sphere (IS) method, which had been refined in a previous study into residential coal combustion (Sun et
al., 2017), was used here to simultaneously quantify BrC and BC. Furthermore, based on this intensive
study of contained biomass burning (in stoves), we extrapolated the results to develop a novel
algorithm for estimating the contribution of solar light absorption by BrC to the sum of BC + BrC for
any combustion source. This will help to gain a clearer idea of whether BC or BrC dominates the light
absorption properties of biomass burning (contained plus open) on a global scale.
**2 Experimental Section**
**2.1 Biomass fuels and stove**

Eleven biomass fuels were tested: they were classified into three groups, i.e. crop residue (CR,

nine types), firewood (FW, one type), and pellet (PF, one types) fuels. The details of these fuels are
given in Table S1. The stove that we used in this study was a natural draft stove developed specifically
for biomass fuels (see Figure S1 in Supporting Information). It is simple and traditional, accounting for
approximately a half of biomass stoves in China (World Bank, China, 2013; Ran et al., 2014).
**2.2 Combustion experiment and sample collection**

The burning and sampling procedures used in this study were in general similar to those described

in a previous coal combustion experiment (Sun et al., 2017). Briefly, each biomass fuel was burned in
the biomass-burning stove. For each biomass fuel, the first batch (30–50 g) was put into the stove and
then ignited with solid alcohol. Sampling and monitoring were immediately initiated. When the
combustion began to fade (the first burning cycle, 3–5 min), a second batch of the fuel was added into
the stove until it had been burned out (the second burning cycle, 3–5 min). Some biomass fuels (e.g.
rice and wheat straws) burned so fast that a third or fourth addition was needed to sustain the
combustion for an adequate sampling period. The modified combustion efficiency (MCE) ranged from





83.95% (peanut stalk) to 99.51% (pellet fuel), with an average of 92.04 ± 4.96%, generally comparable
to the results for residential coal (Average MCE values were 88.0±4.0% and 82.5 ± 17.4% for
bituminous chunk and anthracite chunk, respectively, and were 90.1 ± 1.3% and 92.8 ± 1.7% for all
briquettes tested) (Zhang et al., 2020).

Back to igniting manner, although in most occasions biomass fuels are ignited by gas lighters by

ordinary stove users, there are some difficult-to-ignite biomass fuels (e.g., wood) that need to be
kindled by some flammable soft materials (e.g., wheat straw, rice straw, or even leaves). Additional
emissions from the flammable soft materials are inevitable. In such situations, using solid alcohol to
ignite experimental biomass fuels in this study is appreciated because no pollutants other than $CO_2$ and
$H_2O$ were released from alcohol combustion.

A diversion-dilution-sampling system (Supporting Information, Figure S2) was set up to sample

and/or monitor the combustion emissions. The dilution ratios were 20:1 to 80:1, depending on the
envisaged emission intensity of each combination process, as well as on the burning conditions. The
quartz fibre filters used for sampling were pre-baked in a muffle furnace at 450 °C for 6 h to remove
carbonaceous substances from the filters. Each combustion experiment was repeated 2–3 times to
determine the reproducibility. After sampling, the particle-loaded filters were kept in a freezer at -20 °C
until needed for further analysis.
**2.3 Measurement of BrC with the integrating sphere method**

The differentiation of BrC from BC is a key step toward determining BrC. The mechanism and

procedure of the IS method were detailed in a previous study (Sun et al., 2017). Briefly, a 150 mm IS
(manufactured by Labsphere, Inc, see Figure S3) was built into a UV-Vis-NIR spectrophotometer
(Perkin Elmer Lambda 950). The sphere was internally coated with Polytetrafluoroethylene (PTFE),
which can reflect more than 99% of the incident light in the range of 0.2–2.5 μm (Wonaschüetz et al.,
2009). With this assembly, we scanned through the wavelength range of 350–850 nm to measure the
light absorption of the collected samples.

Two reference materials were used as proxies for BC and BrC. They were carbon black (CarB)

(e.g. Elftex 570, Cabot Corporation) for BC (Fisher, 1970; Andre et al., 1981; Hitzenberger et al., 1996;
Wonaschüetz et al., 2009) and humic acid sodium salt (HASS) (e.g. Acros Organics, no. 68131-04-4)



for BrC (Wonaschüetz et al., 2009). CarB had been used as proxy for BC in diesel exhaust by Medalia
et al. (1983) and HASS had been used as proxy for BrC from wood combustion by Wonaschüetz et al.
(2009). In a previous study, CarB and HASS were used as proxies for BC and BrC, respectively, to
characterise household coal burning samples, by assuming that BC and BrC in household coal
emissions had the same light-absorbing properties as CarB and HASS, respectively (Sun et al., 2017).
In the present study, we continued this logic, and assumed that BC and BrC in household biomass
smoke have the same light-absorbing properties as CarB and HASS, respectively. This approach has
also been adopted in other studies (Heintzenberg, 1982; Reisinger et al., 2008; Wonaschüetz et al., 2009;
Sun et al., 2017). Although such an assumption is not fully perfect, researchers can take advantage of
these two reference materials to relatively assess the features (chemical or optical) of BrC and BC
derived from different combustion sources. It should be noted that the IS method does not depend on an
actual chemical separation, but on a virtual optical allocation of a mixed absorption signal to BrC and
BC, with HASS and CarB used as references, respectively.
Calibration curves (see Figure S4) were plotted for CarB masses from 1.5–90 µg and HASS
masses from 3–240 µg, according to their respective absorption signals as measured by the IS device, at
both 650 nm and 365 nm (Sun et al., 2017). The BrC and BC masses of the samples were calculated
through an iterative procedure based on the different spectral dependences of absorption by BrC and
BC (See Methods for calculation of iteration procedure and Figure S4 in Supporting Information). In
most cases, 20 iterative calculations will achieve a convergent value for either BrC or BC. Note that
carbon accounts only for 47% of the mass of HASS, and therefore all measured HASS equivalent
values based on the calibration curves in Figure S4 were multiplied by 0.47 to obtain the mass of pure
brown 'carbon' (rather than that of the BrC-containing compounds).
**2.4 Calculation methods**
Details of the methods for calculating $EF_{BrC}$, $EF_{BC}$, absorption Ångström exponent (AAE), the
wavelength-dependent BrC contribution to total light absorption ($f_{BrC}(\lambda)$), and average BrC contribution
to total solar light absorption ($F_{BrC}$) in the range of 350–850 nm are provided in the Supporting
Information.
**3 Results and Discussion**



### 3.1 Emission factors of BrC from biomass fuels

The calculated EFs of the 11 biomass fuels are presented in Table 1. $EF_{BrC}$ varied significantly among biomass fuels. Rape straw had the highest $EF_{BrC}$ (7.259 ± 0.002 g/kg), whereas pellet fuel had the lowest (0.13 ± 0.061 g/kg). The observed differences may be related to the type of plant (see Figure 1). We notice that the EFs of BrC for herbaceous plants (HP, the former nine samples in Figure 1) were higher than those for the ligneous plants (LP, the latter two samples in Figure 1). This possibly implies that herbaceous plants have a higher potential for forming BrC than ligneous plants. Although the reason underlying this difference is currently unknown, in view of the higher contents of C and H in LPs than in HPs, it seems reasonable to speculate that burning herbaceous plants in household stoves releases less heat than burning ligneous ones, which leads to a lower burning temperature for the former than for the latter, and therefore favours the generation of BrC for the former (Chen et al., 2015b; Wei et al., 2017). Another possible explanation is the distinction in the modified combustion efficiency (MCE) values between LPs and HPs. Our measurements show that HPs tended to have lower MCEs (93.4 ± 6.49% < 95.9 ± 2.05%), resulting in a greater chance for the formation of BrC (Shen et al., 2013). A similar phenomenon was also observed by Shen et al. (2013), who carried out a systematic measurement of PM, OC, and EC released from various solid fuels burned in residential stoves; these authors found that crop residues, which were composed of herbaceous plants, were more likely to have higher BrC EFs than wood fuels, which were composed of ligneous plants. In this perspective, greater importance ought to be attached to herbaceous biomass fuels than to ligneous ones as far as BrC emissions are concerned.

The $EF_{BC}$ values for PFs were the lowest among all the tested biomass fuels; the briquetting effect helped to lower the occurrence of incomplete combustion and thus likely decreased the formation of primary carbonaceous particles (including BC and BrC) (Zhi et al., 2008, 2009). This agrees with the findings of Lei et al. (2018), as the sum of LAC (BrC + BC) was observed to decrease after the maize straw was transformed to a maize briquette. In view of the virtues of biomass briquetting, regarding both air quality (less pollutant emissions) and climate change mitigation (carbon-neutral), the present study identified an additional benefit of biomass briquetting in climate change mitigation, because of the reduction of the emission of LAC (Sun and Xu, 2012; Arshanitsa et al., 2016; Chen et al., 2016).





Geometrically averaging the $EF_{BrC}$ values over all tested biomass fuels yielded a value of 0.71
g/kg. This value was comparable to the obtained $EF_{BrC}$ for forest fires in the south-eastern United States,
measured with an aethalometer AE52 (1.0–1.4 g/kg, BC-equivalent) (Aurell and Gullett, 2013). In
another study by Schmidl et al. (2008), the IS method was used to measure the BrC and BC emission
characteristics of the open fires of three kinds of leaves. As BrC accounted for 18.5% (w/w) of the
$PM_{10}$ of leaf smoke (Schmidl et al., 2008) and as the $PM_{10}$ EF for biomass fuel combustion (given by
Cao et al. (2011)) is 5.77 g/kg (field burning), the $EF_{BrC}$ can be inferred for the open fires of the three
kinds of leaves, i.e. 1.07 g/kg. This value is also comparable to the averaged $EF_{BrC}$ obtained in this
study. In addition, the current $EF_{BrC}$ average value, 0.71 g/kg, was closer to the values obtained for the
combustion of anthracite-chunks (1.08 ± 0.80 g/kg) and anthracite-briquettes (1.52 ± 0.16 g/kg) than to
those obtained for the combustion of bituminous-chunks (8.59 ± 2.70 g/kg) and bituminous-briquettes
(4.01 ± 2.19 g/kg) (Sun et al., 2017). This suggests the specific importance of the residential
combustion of bituminous coals in BrC emissions.
Figure 1 aids to compare $EF_{BrC}$ and $EF_{BC}$. Each of the 11 biomass fuels tested in this study had a
higher $EF_{BrC}$ than $EF_{BC}$; that is, the ratios of $EF_{BrC}$ to $EF_{BC}$ ($R_{BrC/BC}$) were all >1. Specifically, corncobs
and sorghum stalks give the highest (10.0) and lowest (1.5) $R_{BrC/BC}$ values, respectively, and the
average $R_{BrC/BC}$ over all biomass fuels was 6.7 ± 2.7. This illustrates the significant potential of BrC
emissions than BC emissions, regarding the combustion of household biomass fuel. Kirchstetter et al.
(2004) measured the light absorption of filter-based aerosol samples from biomass burning before and
after acetone treatment (which removed OC). They found that 50% of total light absorption was
attributable to OC. In view of the much smaller average absorption efficiency of BrC, relative to that of
BC, the contribution of BrC to the mass of LAC is undoubtedly far higher than that of BC, an inference
which is consistent with the present study.
**3.2 Spectral dependence of absorption**
AAE represents the spectral dependence of the light absorption efficiency (Martinsson et al., 2015;
Washenfelder et al., 2015; Yan et al., 2015). Usually, the AAE is close to 1.0 (Lack and Langridge,
2013; Laskin et al., 2015) for BC that is pronounced by a graphitic structure. This has been
demonstrated by several studies for diesel exhaust or urban particulate matter (Rosen et al., 1978;



Horvath, 1997). However, the existence of BrC in aerosols makes the mass absorption efficiency
(MAE) increase tend more strongly towards shorter wavelengths, due to a larger AAE for BrC than for
BC. In other words, the AAEs of BrC-containing carbonaceous aerosols are >1 (Chakrabarty et al.,
2013; Yan et al., 2015).

In this study, the measured AAE values for smoke from the combustion of the 11 biomass fuels

(see Table S2) ranged from 1.38 (sorghum stalk) to 2.98 (rice straw), with an average of 2.46 ± 0.53.
This suggests the existence of BrC in the particulate emissions. As a comparison, in a previous study
that used the IS method for household coal combustion (Sun et al., 2017), average AAE values of 2.55
± 0.44 for coal-briquettes and 1.30 ± 0.32 for coal-chunks were obtained (Sun et al., 2017). Cai et al.
(2014) observed an AAE value of 3.02 ± 0.18 for the open burning of wheat straw, and of 1.43 ± 0.26
for household coal burning, using an aethalometer (AE31). Other studies have reported a wide range of
AAE values, dependent on fuels, combustion conditions, aging effects after emission, the wavelengths
covered and the pre-treatment experienced. (see Table S3 in Supporting Information).

However, as AAE >1 for aerosol samples theoretically results from BrC instead of BC

(Martinsson et al., 2015; Washenfelder et al., 2015; Zhi et al., 2015b; Yuan et al., 2016), the wide range
of AAE literature values are believed to be linked to variation in the ratio of BrC to BC ($R_{BrC/BC}$). In
other words, the increase in $R_{BrC/BC}$ theoretically leads to the increase in AAE (Lack and Langridge,
2013). Indirect support for this interpretation can be inferred from existing literature. For example,
Saleh et al. (2014) noticed that the effective absorptivity of organic aerosol in biomass burning
emissions could be parameterised as a function of the ratio of BC to OC (an umbrella term that also
includes BrC). Costabile et al. (2017) found that the AAE (467–660 nm) in the atmosphere of the urban
Po-Valley was positively correlated with the ratio of organic aerosol (OA) to BC ($R^2 = 0.78$), rather
than to OA concentrations alone. The more persuasive scenario concerns WSOC, which is free of BC
($R_{BrC/BC} = +\infty$); for this scenario the AAE reaches its maximum (also see Table S3).

The EFs and AAEs of 11 biomass fuels used in this study and the EFs and AAEs of seven coals

used in a previous study (Sun et al., 2017) are collated and arranged in a scatter plot (Figure 2).
Obviously the AAE values are positively correlated with $R_{BrC/BC}$ values. Considering that the AAE for
pure BC (i.e., $R_{BrC/BC} = 0$) is conventionally accepted as 1.0, we specify the intercept to 1.0 to comply





with the theoretical constraint. The relation between AAE and $R_{BrC/BC}$ can be expressed in Equation (1).

$AAE = 0.199R_{BrC/BC} + 1.00$     $(R^2 = 0.7527)$                    (1)

Equation (1) supports the AAE-$R_{BrC/BC}$ relation in a quantitative way.

**3.3 Light absorption by BrC from household biomass combustion in household stoves**

With the $EF_{BrC}$ and $EF_{BC}$ obtained in the present study, as well as publicly available consumption
data of household biomass fuels, China's BrC and BC emissions from biomass fuels burned in
household stoves can be calculated, following the method described in the Supporting Information. In
2013, the biomass fuels consumed in China comprised 695 Tg (1 Tg = $10^{12}$ g) for household
cooking/heating purposes (Lu et al., 2011; Tian et al., 2011; NBSC, 2014). The calculated BrC
emissions were 712 Gg. South Asia funeral pyres release 92 Gg of BrC in 2011 (calculated with the
double IS system method), which is much less than that from China's household biomass combustion.
This implies a clear need to control BrC emissions from household biomass burning in China.
Figure 3 compares the emissions of BrC and BC from biomass fuels in this study, and from coals
as reported in a previous study (Sun et al., 2017). It is obvious that BrC emissions were always higher
than BC emissions for both household biomass fuels and coals, which is attributable to the higher $EF_{BrC}$
than $EF_{BC}$ for both biomass fuels and coals. It is also interesting to note that, for BrC, biomass fuel
dominated, whereas for BC, coal was more important. This suggests the relative importance of biomass
fuels in controlling BrC.
The calculated huge emissions of BrC for China's household biomass-fuel combustion represent a
strong argument for including BrC in estimating the total light absorption by emissions from burning
biomass. Here, we used $f_{BrC}(\lambda)$ to represent the fraction of BrC absorption in the sum of light absorption
of BrC + BC at individual wavelengths of the scanned spectral ranges (350–850 nm), measured with
the IS. A detailed description of the theory and method for calculating $f_{BrC}(\lambda)$ is given in Supporting
Information. The results of $f_{BrC}(\lambda)$ for biomass fuels in this study are plotted in Figure 4 (blue line).
Evidently, the $f_{BrC}(\lambda)$ increased towards shorter wavelengths: the $f_{BrC}(\lambda)$ at 850 nm was 0.25,
whereas the $f_{BrC}(\lambda)$ at 350 nm increased to 0.8. In addition to the spectrally-dependent $f_{BrC}(\lambda)$ for
biomass fuels, Figure 4 also presents the spectrally dependent $f_{BrC}(\lambda)$ values for coal (red line) as
obtained in a previous study (Sun et al., 2017). The lowest value of $f_{BrC}(\lambda)$ for coal occurred at 0.061



(850 nm), and the highest value occurred at 0.47 (355 nm). The average $f_{BrC}(\lambda)$ for coal was 0.26,
which was distinctly lower than that for biomass fuels. This difference in $f_{BrC}$ between coal and biomass
smoke can be explained by the difference in $R_{BrC/BC}$ between coal and biomass smoke. It is necessary to
exercise caution when attributing the absorption to BrC vs BC based on wavelength dependence
(expressed as AAE). For example, Lack and Langridge (2013) found that the uncertainties in attributed
BrC absorption might be ±33 % when BrC comprised 23% to 41% of total absorption (Assuming an
absorption measurement uncertainty of ±5 %).

Integrating $f_{BrC}(\lambda)$ over the solar spectrum results in $F_{BrC}$, which represents the fraction of solar

radiance absorbed by BrC relative to the total absorption by BC + BrC (refer to the Supplementary
Information for the method for the calculation of $F_{BrC}$). The standard solar spectrum is also plotted in
Figure 4 (yellow line) as a contrast and reference. A value of 0.508 (0.471–0.542) was obtained for the
$F_{BrC}$ of household biomass fuels across the wavelength range of 350–850 nm, which was nearly twice
that of household coal combustion (0.265) in China (Sun et al., 2017).
**3.4 Extrapolation towards a novel algorithm for estimating the relative contribution of BrC**

As $F_{BrC}$ is defined as the ratio of the solar light absorption by BrC to that by (BrC + BC) across

350–850 nm, it is physically dependent on $R_{BrC/BC}$. There is a scarcity of reported $R_{BrC/BC}$ values,
whereas conversely AAE is frequently reported in existing literature. Therefore, the logarithmical
function that can be fitted to the relationship between $R_{BrC/BC}$ and AAE (Figure 2) can be used for the
practical application of expressing $F_{BrC}$ as a function of AAE.

To construct the function for $F_{BrC}$, with AAE as the independent variable, we managed to gather

four pairs of $F_{BrC}$ vs AAE values. Two of these pairs were based on theory. For pure BC (free of BrC),
AAE and $F_{BrC}$ were 1.0 (Lack and Langridge, 2013; Laskin et al., 2015; Yan et al., 2015; Zhang et al.,
2020) and 0.0, respectively; whereas for samples of pure BrC (free of BC), we averaged over the AAE
values in the literature for WSOC or MSOC (free of BC), thus obtaining an AAE value of 6.09 ± 1.25
(Hoffer et al., 2006; Hecobian et al., 2010; Voisin et al., 2012; Srinivas and Sarin, 2013, 2014; Srinivas
et al., 2016; Lei et al., 2018) (Table S3 Part I). The other two pairs of the $F_{BrC}$ vs AAE values were
obtained from our measurements. A previous study (Sun et al., 2017) demonstrated that, when AAE
was 1.58, $F_{BrC}$ was 0.265. In the present study, as mentioned in Section 3.3, an AAE of 2.46 led to an



$F_{BrC}$ of 0.508. These four $F_{BrC}$ vs AAE pairs were used to construct the relationship between $F_{BrC}$ and
AAE (Figure 5). A logarithmical equation was established between $F_{BrC}$ and AAE, with a very high
correlation coefficient.

$F_{BrC} = 0.5519\ln AAE + 0.0067$        (R$^2$ = 0.999)                    (2)

Equation (2) provides a novel algorithm for deriving $F_{BrC}$ from AAE, without consideration of the

process details for any kinds of combustion sources. This helps to broaden insight into biomass burning
issues from contained conditions to open conditions. The results of $F_{BrC}$ for open fresh emissions from
open biomass burning ($F_{BrC-open}$) vary in literature, and most have values below 0.50 (or 50%) (Lack et
al., 2012; Healy et al., 2015; Washenfelder et al., 2015; Srinivas, et al., 2016). We collected AAE$_{-open}$
data from available journal articles and included them in Table S3 (Part II). The calculated average
AAE$_{-open}$ value was 3.44 ± 1.75, which was larger than the AAE$_{-contained}$ value obtained in this study
(2.46 ± 0.53). Substitution of the AAE$_{-open}$ value (3.44 ± 1.75) into Equation (2) leads to a value of
0.685 for $F_{BrC-open}$, which is higher than the $F_{BrC}$ for contained combustion ($F_{BrC\ -contained}$) (0.508),
indicating that BrC's light absorption was more dominant in open biomass burning emissions than in
contained biomass burning emissions.

Assuming that the AAE$_{-contained}$ and AAE$_{-open}$ identified above apply to whole world biomass

burning, we can now assess BrC's role in the biomass burning globally (contained + open) ($F_{BrC-entire}$),
in combination with the respective shares of open and contained burning. Previous studies show that
the annual open and contained biomass burning amounts are 5953 Tg (Wiedinmyer et al., 2011) and
2457 Tg (Fernandes et al., 2007), respectively. This implies that open biomass burning represents 71%
of total biomass burning and contained biomass burning represents 29%. Subsequently, the $F_{BrC-entire}$
can be calculated according to the following equation:

$F_{BrC-entire} = 0.29 \times (0.5519\ln AAE_{-contained} + 0.0067) + 0.71 \times (0.5519\ln AAE_{-open} + 0.0067)$        (2)

With Equation (2), the distribution of $F_{BrC-entire}$ was simulated through the Monte Carlo approach,

as shown in Figure 6. The $F_{BrC-entire}$ was 0.644 on average, and with an 80% probability range it lay
between 0.585–0.699. Particularly, the probability of $F_{BrC-entire}$ being larger than 0.500 was higher than
99%, corroborating the leading role of BrC in the absorption of solar light for total biomass burning
emissions.



## 4 Conclusions

The optical IS approach was used to distinguish BrC from BC in filter samples of the emissions of 11 types of biomass after burning in a typical stove. The measured average EF of household biomass fuels for BrC was 0.71 g/kg, and the calculated annual BrC emissions from China's household biomass burning amounted to 712 Gg. This is higher than the emissions from China's household coal combustion (592 Gg). Moreover, it was observed that BrC contributed to approximately half of all light absorption by BC + BrC across the strongest solar spectral range (350–850 nm; $F_{BrC}$ = 50.8%). Furthermore, a novel relationship was constructed ($F_{BrC}$ = 0.5519lnAAE + 0.0067, $R^2$ = 0.999), which can simplify the calculation of $F_{BrC}$ by using AAE. With this mathematical relationship, we calculated the $F_{BrC}$ values for open biomass burning ($F_{BrC-open}$ = 70.1%) and entire biomass burning ($F_{BrC-entire}$ = 64.4%), thereby establishing the dominant role of BrC in biomass burning absorption. From this perspective, we recommend that it is necessary to include BrC in the climate discussion, particularly concerning biomass burning (contained and open). This algorithm omits the long procedures of chemical treatment, optical measurement and tedious calculations, and provides a scheme for estimating the contribution of BrC relative to BC in any combustion process with LAC emissions.

## Data availability

The research data can be accessed, on request, from the corresponding author (zhigr@craes.org.cn).

## Acknowledgements

This study was supported by the National Natural Science Foundation (41977309), Research results of 13[th] five-year plan for Social Sciences in Jiangxi Province, China (19ZK34), National Key Research & Development Plan (2017YFC0213001), Special Project of Fundamental Research Funds of the Chinese Research Academy of Environmental Sciences (JY41373131), Chinese-Norwegian Project on Emission, Impact, and Control Policy for Black Carbon and its Co-benefits in Northern China (CHN-2148-19/0029), and Topics of Jiangxi Sports Bureau, China (2018021).



*Competing interests.* The authors declare that they have no conflicts of interest.





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





**Table 1. Measured EF$_{BrC}$ and EF$_{BC}$ (g/kg) values for household biomass burning**

| Biomass fuels | EF$_{BrC}$ | EF$_{BC}$ |
|---|---|---|
| Rape straw | 7.259 ± 0.002 | 2.537 ± 0.001 |
| Rice straw | 2.50 ± 3.064 | 0.31 ± 0.25 |
| Wheat straw | 1.25 ± 0.074 | 0.13 ± 0.039 |
| Cotton straw | 0.89 ± 0.51 | 0.10 ± 0.019 |
| Bean straw | 0.57 ± 0.12 | 0.089 ± 0.035 |
| Corncob | 0.56 ± 0.55 | 0.056 ± 0.017 |
| Peanut stalk | 0.54 ± 0.15 | 0.13 ± 0.054 |
| Sorghum stalk | 0.45 ± 0.32 | 0.30 ± 0.054 |
| Maize straw | 0.45 ± 0.76 | 0.053 ± 0.014 |
| Pine | 0.27 ± 0.29 | 0.034 ± 0.017 |
| Pellet fuels | 0.13 ± 0.061 | 0.023 ± 0.037 |
| Geomean | 0.71 (0.24, 2.18) | 0.12 (0.033, 0.438) |

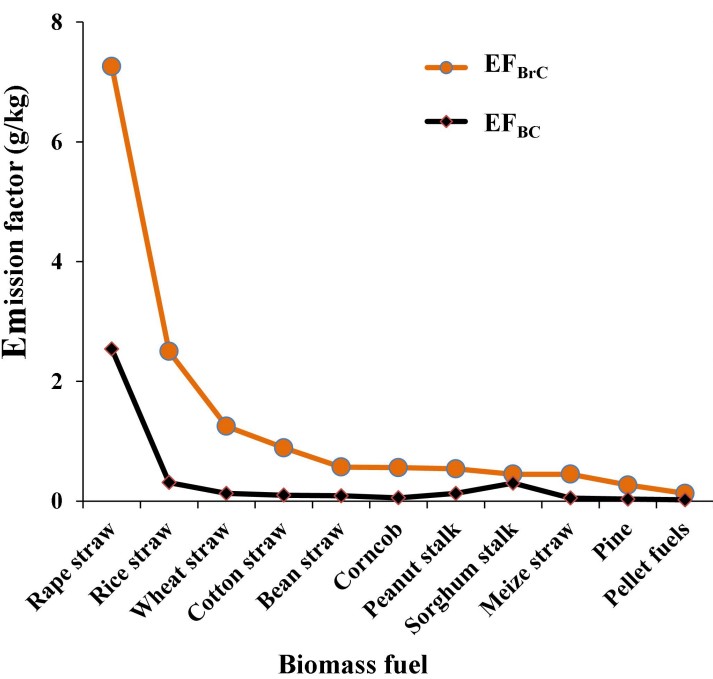

**Figure 1**. EFs of tested biomass fuels

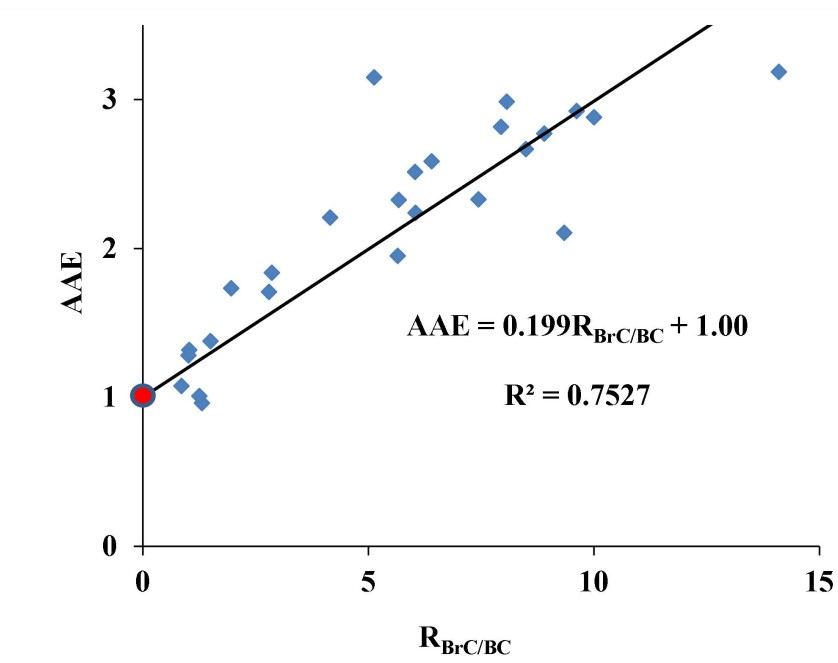

$$AAE = 0.199R_{BrC/BC} + 1.00$$

$$R^2 = 0.7527$$

**Figure 2. Relationship between AAE and EF$_{BrC}$/EF$_{BC}$ ratio (R$_{BrC/BC}$) for both biomass fuel and**

**coal.** The intercept is designated as 1.0 to echo the conventionally accepted notion that the

5          AAE for pure BC (i.e., R$_{BrC/BC}$ = 0) is 1.0.

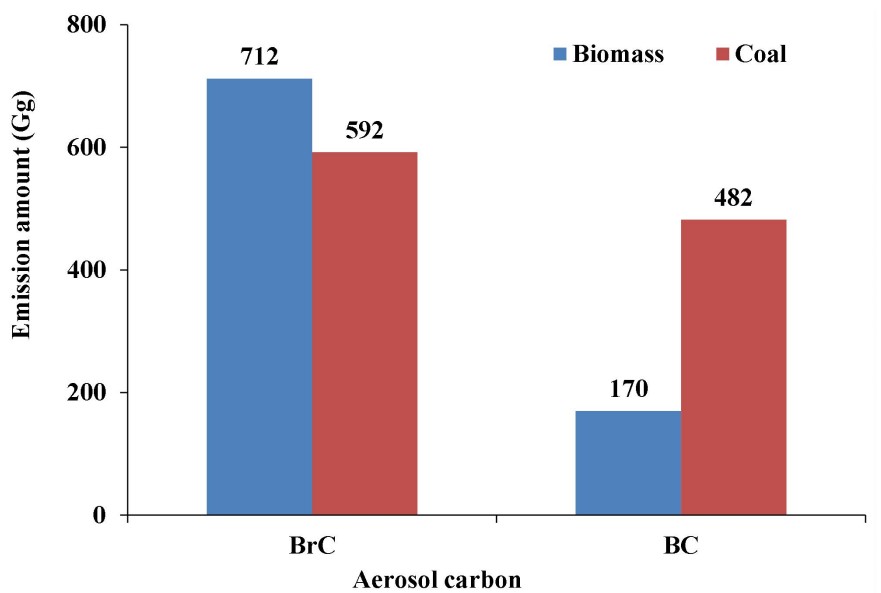

**Figure 3. Comparison of BrC and BC emissions between biomass burning and coal combustion**

**in China's household sector of 2013**





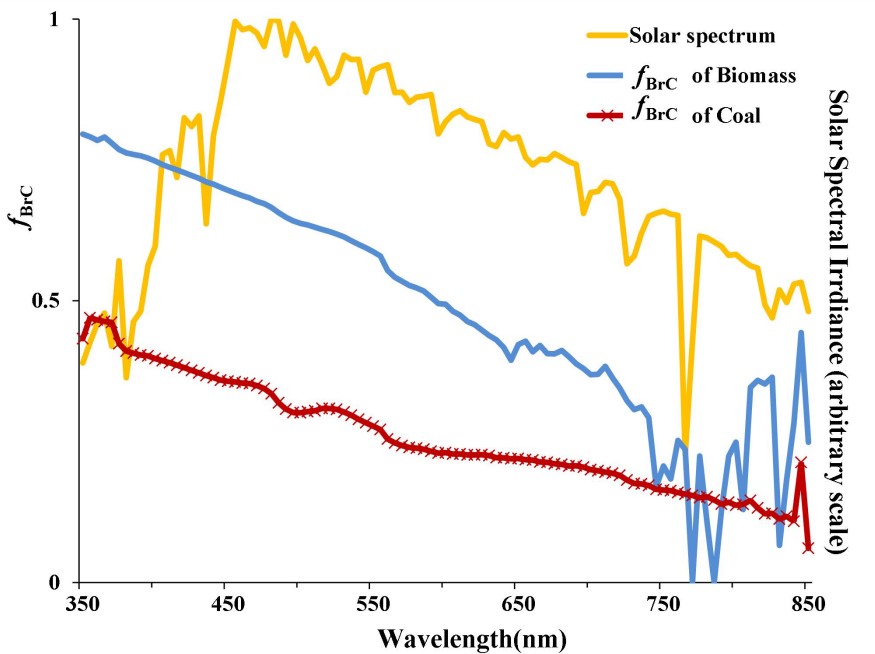

**Figure 4. Ratios of light absorption by BrC to total absorption by total mass with respect to China's household biomass and coal burning**

Note: The ratio is expressed as $f_{BrC}$ and was calculated in accordance with the method described in the Supporting Information. The yellow line is the clear sky global horizontal solar spectrum at the earth's surface for one optical air mass in relative units (Levinson et al., 2010; Chakrabarty et al., 2014)



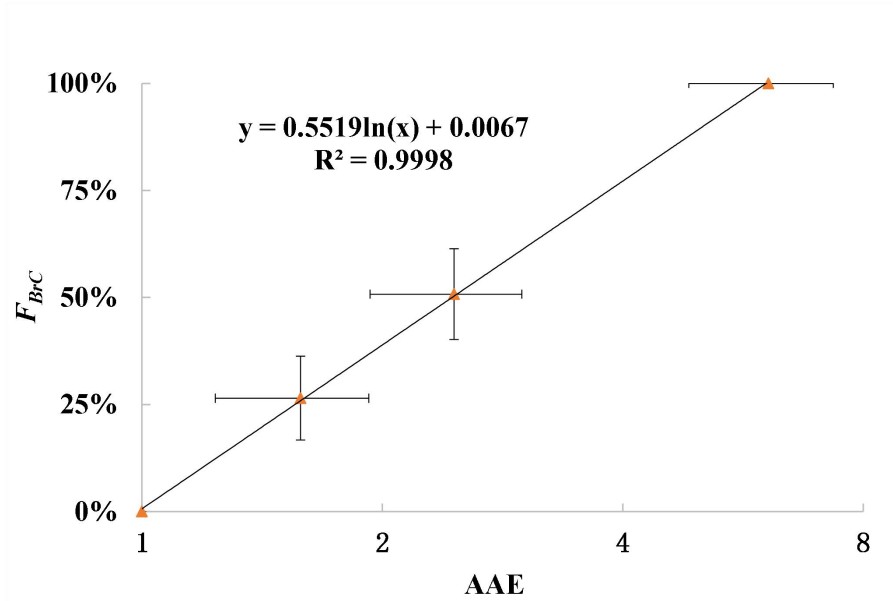

**Figure 5. Relationship between $F_{BrC}$ and AAE**

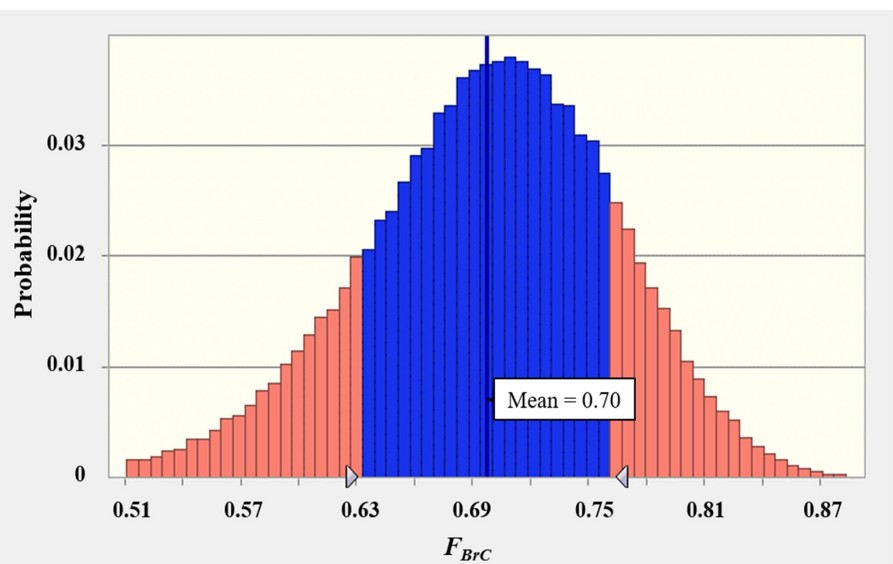

**Figure 6. The probability distribution of calculated $F_{BrC\text{-entire}}$.** Assuming the AAE$_{\text{-contained}}$ value of

$2.46 \pm 0.16$ (mean $\pm$ SD of the means) and AAE$_{\text{-open}}$ value of $3.44 \pm 0.42$ (mean $\pm$ SD of the means)

apply to whole world biomass burning, the combined value for entire biomass burning ($F_{BrC\text{-entire}}$) can

5     be calculated as: $F_{BrC\text{-entire}} = 0.71 \times (0.5519 \ln AAE_{\text{-open}} + 0.0067) + 0.29 \times (0.5519 \ln AAE_{\text{-contained}} +$

$0.0067)$