# Peer review of "Brown carbon's emission factors and optical characteristics in household biomass burning: Developing a novel algorithm for estimating the contribution of brown carbon"

_Atmospheric Chemistry and Physics, 2020_

## Referee Comment (RC1) · Anonymous Referee #1 · 13 Aug 2020

**Title: Brown carbon's emission factors and optical characteristics in household biomass burning: Developing a novel algorithm for estimating the contribution of brown carbon**

This is a manuscript that reports on the emission factors and optical characteristics of BB-derived BrC and development of a novel algorithm for estimating the contribution of BrC. The results indicated the mean emission factors of BB-BrC are 0.71 g/kg, which were affected by the plant type and burning styles. The average AAE value was $2.46 \pm 0.53$, which are much higher than that of coal-chunks combustion smoke. The contribution of absorption by BB-BrC to the total absorption by BC + BrC were also calculated, is 50.8%. Finally, a novel algorithm was developed for estimating the $F_{BrC}$ for any combustion sources. This is an interesting research about the emission factors and light-absorption characteristics of BrC emitted from biomass burning. I think the manuscript can be accepted after the following comments are addressed.

Comments:
1) Line 11: what's the meaning of "0.24, 2.18"?
2) Lines 70-76: several important references for the BrC from biomass burning in China were missed, such as Fan et al. (2016) ACP, 16, 13321-13340; Huo et al. (2018) Atmos. Environ., 191, 490-499, etc.
3) Experimental section: accuracy, precision, and repeatability are not well quantified or discussed in this paper. The 11 biomass fuels are each burned and sampled once. The filter sample for each fire is collected in background air, so ambient aerosol may present in the sample. These may be reasonable experimental procedures, but the following information is missing: i) Blank filter sample for ambient air only to determine the background concentrations; ii) Repetitions of identical sample burns to determine the repeatability of the fires and the analysis procedure.
4) Please reduce the number of significant digits (2-3 is preferred) in Table 1, S1,

and possibly in the main text. For example, "7.259" (four significant digits) can be present as "7.26" (maximum three significant digits). Please double check such errors throughout the entire manuscript.

5) Lines 205-208: The ratios of $EF_{BrC}$ to $EF_{BC}$ for different samples were varied with very large range (the highest one is 10.0 and the lowest one is 1.5). Why? Please add some explanation. This is very important for the estimation of the contribution of BB BrC.

6) Figure 2: the data of BrC from BB and coal combustion should be label with different markers.

7) Lines 252-258: China's BrC and BC emissions from biomass fuels burned in household stoves were calculated. This section is associated with high uncertainties due to the reliable consumption amounts of different types of biomass fuels and forms, representative BrC emission factors from this study. I'd like to suggest to add discussions on uncertainties and limitations.

8) Section 3.4, Lines 295-306: To construct the function for $F_{BrC}$, with AAE as the independent variable, four pairs of $F_{BrC}$ vs AAE values were investigated: one pure BC and three pure BrC. For the three pure BrC, I have two questions: 1) why the average values of $F_{BrC}$ vs AAE rather than the data of each sample were used to construct the function between $F_{BrC}$ and AAE? 2) As shown in Table S3, the AAE values of WSOC or MSOC in the literature were determined in solution. However the AAE values of BrC were determined with the integrating sphere method in this paper and the previous study (Sun et al., 2017). How about the differences of AAE values measured with these two methods. You should add some discussions to interpret that.

9) Table S1: The abbreviation of "M%, CR, FW, PF" should be illustrated in full name.

10) Is Fig S4 cited from the literature of authors (Sun, J., Zhi, G., et al., Emission factors and light absorption properties of brown carbon from household coal combustion in China, Atmos. Chem. Phys., 17, 4769-4780)? If so, please add references in the caption.

---

## Referee Comment (RC2) · Anonymous Referee #2 · 25 Aug 2020

This paper presents emission factors for brown carbon and black carbon from 11 different biomass fuels in a commonly used cookstove. Most of the paper is focused on the development of an algorithm to convert AAE into the mass ratio of BrC to BC and solar absorption fraction attributed to BrC. The paper makes the important point that BrC absorption needs to be included in assessments of the climate impacts from biomass burning. Given the data presented here (one stove and several biomass fuels), the universality of the algorithm for multiple emission sources is overstated. Also, there is limited comparison of the algorithm with other methods, so it is not clear if it is

an improvement over other approaches of estimating the impact of BrC on climate. I recommend major revisions to address the following comments:

General Comments:

1. The quantification of BrC as a mass emission is a relatively uncommon approach due to the complexity of BrC (i.e. many different chromophores with differing mass absorption efficiencies that are source dependent). A fuller description of the proxies used in this current study and how they compare to other black and brown carbon sources should be outlined (e.g. absorption efficiencies, AAE, primary particle size, etc.) and any shortcomings of using these proxies should be noted. How does this method of estimating BrC mass emissions compare with other approaches used in the literature that were cited for comparison of BrC emission factors (e.g. Aurell and Gullett 2013 and Schmidl et al. 2008)?

2. More description of the test protocol is needed, e.g. cold start? size of fuel? How was it determined that the test method was relevant for real-world stove emissions?

3. Overall, there needs to be an analysis of the uncertainty or the error and potential impacts of the assumptions in the algorithm? What is the impact of assuming AAE = 1 for BC? How might lensing impact this analysis? What is the impact of measurement limit of detection?

4. How does the novel algorithm presented here compare with other approaches to quantifying the fractional contribution of BrC and BC to absorption? Is there a benefit to calculating a BrC mass emission factor over other approaches based on AAE? A few studies that may pertain might be Corbin et al. 2018 (https://doi.org/10.1029/2017JD027818), Tian et al. 2019 (https://doi.org/10.1029/2018JD029352), or Zhang et al. 2018 (https://aaqr.org/articles/aaqr-17-12-ac3-0566.pdf) among many others.

5. There is no validation that this algorithm works for sources other than the cookstove

samples measured in this study and in Sun et al. 2017. Unless the authors can include some additional data points from some other sources in their algorithm development the statements made throughout the paper about the wide applicability of the algorithm for 'any combustion sources' are unsupported and should be removed.

6. The manuscript needs to be edited for language, see minor comments for specific examples. But generally, if you are writing 'in other words' it means your first explanation should be simplified and stated only once.

Specific Comments:

Line 30-31: The sentence needs to be rewritten since the two clauses of this sentence are saying the same thing, BrC absorbs more at shorter wavelengths.

Line 46: Why are the units in mg/m2? Most species in the atmosphere are reported in terms of concentration. Is this a typo?

Line 71: Need to be clearer about what characteristics are being referred to here. There are many references published on emissions which measure chemical composition, size distribution, and some even quantify optical properties. Most do not report a BrC emission factor because there is no standard for quantifying BrC mass.

Line 114: These 'soft materials' are usually referred to as kindling and is commonly used when igniting wood and the emissions from a kindling ignition should be included in this analysis since they are representative of real-world use.

Line 120: What is meant by 'envisaged emission intensity'? How was this determined? Is this just the concentration in the sampling duct?

Line 172: Please cite a reference and quantify how much lower the burning temperature or heat release is for herbaceous fuels to support this speculation.

Lines 175 – 181: Were no other measurements made during the tests (e.g. CO, CO2, PM, EC, OC)? These other measurements would greatly support some of the specu-

lation in this section. I am not sure the speculation is justified without measurements from actual study here.

Lines 205: This paragraph needs to be revised for language usage.

Line 208: What is meant by 'the significant potential of BrC emissions than BC emissions'? Does this mean larger emissions? Larger mass fractions? Larger BrC/BC ratios? Larger impact? Be specific about what quantity is of BrC emissions is significant and by what amount.

Line 212: Please provide the average absorption efficiencies of BrC and BC that are being referenced for this statement.

Line 255-57: Why are funeral pyres used as an emissions comparison? It seems like an odd source to include and to leave out any mention of open burning (e.g. ag residues, forest fires) or coal for cookstoves. Is coal included in the 'biomass fuels' in mentioned in line 255?

Line 280: What was the source of the uncertainties in the Lack and Langridge analysis? Do they apply in this study?

Line 328-331: How does this compare to the direct radiative forcing attributed to BrC referred to in the introduction?

Figure 1: Please include error bars to show the uncertainty in the measurement. Presumabley repeat measurements were made because there are standard deviations (standard error?) provided in Table 1.

Table 1: Please include all the quantities measured and calculated for each sample (e.g. AAE, RBrC/BC, fBrC, FBrc) along with propagated uncertainties.

Figure 4: What is the impact of limit of detection on this plot? The data > 750 nm is very noisy, and I wonder if that is not due to limitations of the measurement? If this data is below the limit of detection it should not be used in the calculation of fBrC.

Figure 5 and Line 303: Why only use the mean (median?) fBrC from these current study and Sun et al. 2017? Although the regression is strongly correlated here, the scatter in the data is covered up by using the mean value instead of every measured data point.

SI:

Tables S3 Part I: Extracts are dominated by ambient aerosols, what about source? E.g. fossil fuel combustion, woodstoves, open burning? (Just a few examples are: Xie et al. 2017 https://doi.org/10.1038/s41598-017-06981-8 for open burning and gasoline exhaust; Xie et al. 2018 https://doi.org/10.1016/j.envpol.2018.04.085 for cookstoves using wood, kerosene and charcoal; Corbin et al. 2018 https://doi.org/10.1029/2017JD027818 for marine diesel engines). Since this paper is focused on emissions it would be good to have a more exhaust list of emissions AAE measurements. Calculations: should 'coal' here be 'biomass fuel'?

Figures S2-S4: Appear to be identical to those in Sun et al. 2017, should the reference be noted in the caption?

Figure S4: Hard to follow the text here, would be easier to understand in equation form or even a diagram.

---

## Author Comment (AC1) · 29 Oct 2020

The comment was uploaded in the form of a supplement:
https://acp.copernicus.org/preprints/acp-2020-548/acp-2020-548-AC1-supplement.zip

---

## Author Comment (AC2) · 29 Oct 2020

The comment was uploaded in the form of a supplement:
https://acp.copernicus.org/preprints/acp-2020-548/acp-2020-548-AC2-supplement.zip

---

## Author Response (AR1)

J. Sun et al.

zhxi.2006@163.com

**Reply to Referee 1**

**General comment:**

This is a manuscript that reports on the emission factors and optical characteristics of BB-derived BrC and development of a novel algorithm for estimating the contribution of BrC. The results indicated the mean emission factors of

BB-BrC are 0.71 g/kg, which were affected by the plant type and burning styles. The average AAE value was $2.46 \pm 0.53$, which are much higher than that of coal-chunks combustion smoke. The contribution of absorption by BB-BrC to the total absorption by BC + BrC were also calculated, is 50.8%. Finally, a novel algorithm was developed for estimating the $F_{BrC}$ for any combustion sources. This is an interesting research about the emission factors and light-absorption characteristics of BrC emitted from biomass burning. I think the manuscript can be accepted after the following comments are addressed.

**Response:**

Thanks a lot for the positive comment. The recommendation for publication in ACP

is encouraging. We would further improve our manuscript according to the comments and suggestions below.

**Comments 1:**

Line 11: what's the meaning of "0.24, 2.18"?

**Response:**

Thanks. When we initially submitted our manuscript to editorial office, the editor suggested us not to use arithmetic mean but other forms, such as the median value and/or a range because the values of $EF_{BrC}$ (or $EF_{BC)}$ for different samples differed by more than an order of magnitude. Following the suggestion, we turned to use geomeans instead of arithmetic means. Line 11 in original version described the geomean (i.e., 0.71g/kg) calculated for $EF_{BrC}$ and the (lower, upper) limits (i.e., 0.24, 2.18) calculated via a geomean (i.e., 0.71g/kg) divided/multiplied by the geometric standard deviation (i.e., 2.95). We added a note to Table 1 to show how the geomean and range (lower and upper limits) were obtained. In addition, we changed the upper limit of 2.18 in original version to 2.09 in revised version to correct for wrong calculation in original version. In line 11 (revised version), we changed 0.71 g/kg (0.24, 2.18) to 0.71 g/kg (0.24-2.09).

**Comments 2:**

Lines 70-76: several important references for the BrC from biomass burning in China were missed, such as Fan et al. (2016) ACP, 16, 13321-13340; Huo et al. (2018) Atmos. Environ., 191, 490-499, etc.

**Response:**

Thanks for this reminder. We added these citations proposed by the reviewer (revised version, lines 72-73).

**Comments 3:**

Experimental section: accuracy, precision, and repeatability are not well quantified or discussed in this paper. The 11 biomass fuels are each burned and sampled once. The filter sample for each fire is collected in background air, so ambient aerosol may present in the sample. These may be reasonable experimental procedures, but the following information is missing: i) Blank filter sample for ambient air only to determine the background concentrations; ii) Repetitions of identical sample burns to determine the repeatability of the fires and the analysis procedure.

**Response:**

So sorry that we didn't completely and clearly describe the accuracy, precision, and repeatability. In the revised version, we added the missing information and bettered the unclear description, particularly on the two key concerns: i) Blank filter sample for ambient air only to determine the background concentrations; ii) Repetitions of identical sample burns to determine the repeatability of the fires and the analysis procedure.

The revised version is as follows: "Each of the 11 biomass fuels was burned for 2-3 times and the emissions were collected on individual filters. The 2-3 duplicate samples helped check the reproducibility and analysis procedure. Background concentrations in ambient air were obtained separately." (lines 107-109, revised version).

**Comments 4:**

Please reduce the number of significant digits (2-3 is preferred) in Table 1, S1, and possibly in the main text. For example, "7.259" (four significant digits) can be present as "7.26" (maximum three significant digits). Please double check such errors throughout the entire manuscript.

**Response:**

Thanks for this reminder. We checked throughout the whole manuscript and reduced the number of significant digits after the decimal point to 2 in Table 1 and Table S1 except for $EF_{BrC}$ and $EF_{BC}$ of the rape straw. We actually collected 2 samples for rape straw. The experimental results of the duplicates were extremely close, which made the standard deviations be 0.002 for $EF_{BrC}$ and 0.001 for $EF_{BC}$. When we initially submitted our manuscript to editorial office, the editor suggested us to increase the number of significant digit after the decimal point from 2 (0.00) to 3 (0.002 and 0.001 specifically for the $EF_{BrC}$ and $EF_{BC}$ of rape straw sample) to avoid the uncertainty value of 0.00. For this reason, we maintained the two data of rape straw with 3 significant digits after the decimal point, as 7.259±0.002 and 2.537±0.001 (See revised version, Table 1 and S1), while the data of other biomass fuels were designated with 2 significant digits after the decimal point.

**Comments 5:**

Lines 205-208: The ratios of $EF_{BrC}$ to $EF_{BC}$ for different samples were varied with very large range (the highest one is 10.0 and the lowest one is 1.5). Why? Please add some explanation. This is very important for the estimation of the contribution of BB BrC.

**Response:**

Thanks for this suggestion. Sure the ratios of $EF_{BrC}$ to $EF_{BC}$ for different samples varied with very large ranges; for example the highest one is 10.0 and the lowest one is 1.5. Although the reasons for the large range in $R_{BrC/BC}$ ratio among different biomass cases involves very complicated factors, they are essentially attributed to the differences in chemical composition and physical structure. It is acknowledged that both BrC and BC are products of incomplete combustion of biomass fuels (Andreae and Gelencsér, 2006; Yan et al., 2015). Different biomass fuels are composed by different organics that have different combustion performances (Reid et al., 2005; Saleh et al., 2014); meanwhile, different biomass fuels are also different in densities and moistures (Shen et al., 2014; Jacobson et al., 2015), which are also potential to exert influences on the combustion performance. The combustion performance relates to something like the combustion speed and temperature, both of which are important to the formation of BrC and BC. Usually a low combustion temperature is more favorable for BrC formation and a relatively high combustion temperature is more favorable for BC formation (Chen and Bond, 2010; Bond et al., 2013; Shen et al., 2014). This suggests that the generation processes of BC and BrC are often not synchronous but in opposite trend, which made the values of $R_{BrC/BC}$ vary terribly.

We understand the importance of the $R_{BrC/BC}$ for the estimation of the contribution of biomass BrC and accordingly added some explanations in lines 221-235 in our revised version.

**Comments 6:**

Figure 2: the data of BrC from BB and coal combustion should be labelled with different markers.

**Response:**

Thank you. In our revised version, the data of BrC from BB and from coal combustion in Fig. 2 have been labelled with red and blue markers, respectively.

**Comments 7:**

Lines 252-258: China's BrC and BC emissions from biomass fuels burned in household stoves were calculated. This section is associated with high uncertainties due to the reliable consumption amounts of different types of biomass fuels and forms, representative BrC emission factors from this study. I'd like to suggest to add discussions on uncertainties and limitations.

**Response:**

Thanks for this suggestion. Lines 252-258 (previous version) described China's

BrC and BC emissions from biomass fuels burned in household stoves. The calculated emissions indeed contained uncertainties resulting from the consumption amounts and forms of different types of biomass fuels as well as the representativity of BrC

emission factors measured in this study. We added discussions on the uncertainties and limitations (lines 286-289, revised version).

**Comments 8:**

Section 3.4, Lines 295-306: To construct the function for $F_{BrC}$, with AAE as the independent variable, four pairs of $F_{BrC}$ vs AAE values were investigated: one pure

BC and three pure BrC. For the three pure BrC, I have two questions: 1) why the average values of $F_{BrC}$ vs AAE rather than the data of each sample were used to construct the function between $F_{BrC}$ and AAE? 2) As shown in Table S3, the AAE

values of WSOC or MSOC in the literature were determined in solution. However the

AAE values of BrC were determined with the integrating sphere method in this paper and the previous study (Sun et al., 2017). How about the differences of AAE values measured with these two methods. You should add some discussions to interpret that.

**Response:**

Thanks for this question and suggestion.

(A) We described how we constructed the relation in lines 330-341 (revised version). That is, to construct the function for $F_{BrC}$, with AAE as the independent variable, we managed to gather four pairs of $F_{BrC}$ vs AAE values. Two of these pairs were theoretically for pure BC and pure BrC, respectively. For pure BC (free of BrC), the AAE and $F_{BrC}$ were 1.0 (Lack and Langridge, 2013; Laskin et al., 2015; Yan et al.,

2015; Zhang et al., 2020) and 0.0, respectively, and for samples of pure BrC (free of

BC), we averaged over the AAE values in the literature for WSOC or MSOC, thus obtaining an AAE value of 6.09 ± 1.45 (Hoffer et al., 2006; Hecobian et al., 2010;

Voisin et al., 2012; Srinivas and Sarin, 2013, 2014; Srinivas et al., 2016; Lei et al.,

2018) (Part I in Table S3). The other two pairs of the $F_{BrC}$ vs AAE values were obtained from our previous and current measurements. The previous study (Sun et al.,

2017) demonstrated that, when AAE was 1.58, $F_{BrC}$ was 0.265. In the present study, as mentioned in Section 3.3, an AAE of 2.46 led to an $F_{BrC}$ of 0.508. These four $F_{BrC}$

vs AAE pairs were used to construct the relationship between $F_{BrC}$ and AAE (Figure

5).

(B) The question why the average values of $F_{BrC}$ vs AAE rather than the data of each sample were used to construct the function between $F_{BrC}$ and AAE is worth explaining.

The same question had actually been raised by the editor and we had explained the reason in advance. On the one hand, we know, each of the latter three points (i.e., 1.58,

0.265; 2.46, 0.508; 6.09, 1.00) in Figure 5 is the average of a number of data, and therefore each of them can be potentially replaced with a cluster of individual dots if we like; yet on the other hand, the first point (0.00,1.00) is not originated from averaging over a cluster of individuals but from theoretical consideration, and thus there are no cluster of individual dots usable to replace this single point. Under the circumstances, replacing each of the latter three points with an individual cluster of dots while leaving the first point with single dot will substantially lower the weight of the first point from 25% to almost being negligible. Given the theoretical significance of the first point, this is not only unfair but also unacceptable. For this consideration, we preferred to the average value for each of the latter three points so that all the four points in Figure 5 were put weights equally (25%). Additionally, compared with a cluster of individuals, an average is usually closer to or more representative of the true value and hence is more persuasive. We added an explanation in our revised version (lines 340-341).

(C) As regards the need to add some discussion on the differences between AAE

values measured with IS method and the AAE values measured through WSOC or

MSOC, the former is for the entirety of a sample including BrC+BC whereas the latter is for BrC alone (free of BC).

**Comments 9:**

Table S1: The abbreviation of "M%, CR, FW, PF" should be illustrated in full name.

**Response:**

Thanks for reminder. We gave the full names of the abbreviations as a note to Table

S1, as follows:

Note: M% - moisture on air-dry basis (%); 11 biomass fuels used in this study were divided into 3 categories: CR - crop residue; FW - fire wood; PF - pellet fuel.

**Comments 10:**

Is Fig S4 cited from the literature of authors (Sun, J., Zhi, G., et al., Emission factors and light absorption properties of brown carbon from household coal combustion in China, Atmos. Chem. Phys., 17, 4769-4780)? If so, please add references in the caption.

**Response:**

Thanks for reminder. We added 'Sun et al., 2017' in Fig. S4 accordingly.

Line 114: These 'soft materials' are usually referred to as kindling and is commonly used when igniting wood and the emissions from a kindling ignition should be included in this analysis since they are representative of real-world use.

**Response:**

Thanks for this comment. The paragraph (lines 114-118 in original version and lines 115-120 in revised version) intends to show: (i) in ordinary practice, there are some difficult-to-ignite biomass fuels (e.g., wood) that need to be kindled by some flammable soft materials (e.g., wheat straw, rice straw, or even leaves) and therefore additional emissions from the flammable soft materials must be considered; (ii) however in our study, only solid alcohol was used to ignite experimental biomass fuels and almost no pollutants other than $CO_2$ and $H_2O$ were released from alcohol combustion.

Line 120: What is meant by 'envisaged emission intensity'? How was this determined? Is this just the concentration in the sampling duct?

**Response:**

Thanks for this question. The emission intensities of different biomass fuels varied greatly, so we have to properly set appropriate dilution ratios for different biomass fuels to meet the experimental needs. The 'envisaged emission intensity' was obtained from two approaches, one from our experiences in household solid fuel combustion experiments, and the other from sufficient pre-experiments.

A stream of flue gas was ducted from the stovepipe into the diluter. That is, the concentration before the diluter was the same as in the stovepipe and the concentration after the diluter was lower than in the stovepipe. The 'envisaged emission intensity' mentioned in this study refers to the concentration inside the stovepipe or before the diluter. The dilution ratios were preset depending on the envisaged emission intensity. Please see the description in lines 119-125.

Line 172: Please cite a reference and quantify how much lower the burning temperature or heat release is for herbaceous fuels to support this speculation.

**Response:**

Thanks for this suggestion. In lines 189-191, we added a sentence "In this study, the temperature tested in the stovepipe (50 cm above the stove upper surface) for HPs was 62.9 °C

while for LPs, was 77.1 °C ".

Lines 175 – 181: Were no other measurements made during the tests (e.g. CO, $CO_2$, PM, EC, OC)? These other measurements would greatly support some of the speculation in this section. I am not sure the speculation is justified without measurements from actual study here.

**Response:**

Thanks for the suggestion. We do have got some data during the tests, including organic carbon (OC), elemental carbon (EC), and modified combustion efficiency (MCE) of every combustion experiment (Table S4 here). OC and EC values were extracted from our previous publication (Sun et al., 2018). These data favor our speculation mentioned in this section.

Table S4 The values of MCEs of every samples

| Sample ID | Biomass fuels | MCE (%) | $EF_{OC}$ (g/kg) | $EF_{EC}$ (g/kg) |
|---|---|---|---|---|
| 1 | rape straw | 88.12 | 15.46 | 3.43 |
| 2 | peanut stalk | 83.95 | 0.53 | 0.05 |
| 3 | rice straw | 93.40 | 2.76 | 0.35 |
| 4 | wheat straw | 84.83 | 0.82 | 0.10 |
| 5 | bean straw | 92.70 | 0.67 | 0.081 |
| 6 | corncob | 99.21 | 1.15 | 0.12 |
| 7 | sorghum stalk | ~100.00 | 0.28 | 0.08 |
| 8 | maize straw | 99.86 | 0.76 | 0.086 |
| 9 | cotton straw | 98.63 | 0.91 | 0.16 |
| 10 | pine | 97.34 | 0.37 | 0.063 |
| 11 | pellet fuel | 94.45 | 0.05 | 0.016 |
| | Mean | 93.86 | 2.16 | 0.42 |

Lines 205: This paragraph needs to be revised for language usage.

**Response:**

Thanks for this comment. We have carefully read this paragraph and have tried to improve the language. Particularly, the next two comments of this reviewer and a comment of another reviewer are all regarding this paragraph and have incurred immense changes in the text. The paragraph in our original version has now even been expanded into two paragraphs (lines 221-235 and lines 236-243). We paid great attention to language usage when constructing these two paragraphs.

Line 208: What is meant by 'the significant potential of BrC emissions than BC emissions'? Does this mean larger emissions? Larger mass fractions? Larger BrC/BC ratios? Larger impact? Be specific about what quantity is of BrC emissions is significant and by what amount.

**Response:**

Sorry for the ambiguity. This sentence has been deleted in our revised version without affecting our intent.

Line 212: Please provide the average absorption efficiencies of BrC and BC that are being referenced for this statement.

**Response:**

Thanks. We have provided a set of MAE values for BC, BrC, and dust in lines 241-242 (revised version) to show the huge difference between the MAEs of BC and BrC. In Yang et al.(2009), the MAEs at 550 nm were estimated to be 9.5, 0.5, and 0.03 $m^2/g$, respectively, for BC, BrC, and dust.

Line 255-57: Why are funeral pyres used as an emissions comparison? It seems like an odd source to include and to leave out any mention of open burning (e.g. ag residues, forest fires) or coal for cookstoves. Is coal included in the 'biomass fuels' in mentioned in line 255?

**Response:**

Indeed, funeral pyres combustion is an odd source for comparison. However the studies regarding the emission factors of biomass BrC were so scarce, we had to mention funeral pyres combustion as one source of information.

In line 255 (original version), coal is not included in the 'biomass fuels' (line 284 in revised version).

Line280: What was the source of the uncertainties in the Lack and Langridge analysis? Do they apply in this study?

**Response:**

The uncertainty analysis in Lack and Langridge (2013) includes the uncertainty of AAE allocation method (the uncertainty of AAE of BC) and the uncertainty of experiment (the uncertainty of instrument measurement). We quoted them here just for knowing the potential of uncertainty subject to AAE.

Line 328-331: How does this compare to the direct radiative forcing attributed to BrC referred to in the introduction?

**Response:**

Thanks for this comment. The $F_{BrC}$ and radiative forcing (RF) are of different concepts. The former refers to "the contribution of absorption by BrC to the total absorption by BC + BrC across the strongest solar spectral range of 350–850 nm" (see lines 16-17 in Abstract), while the latter refers to the difference of insolation (sunlight) absorbed by the Earth and energy radiated back to space (https://encyclopedia.thefreedictionary.com/Radiative+forcing). There is no fixed relation between them. However, the knowledge of $F_{BrC}$ helps identify which one of BC and BrC dominates the light absorption of solar radiation.

Figure 1: Please include error bars to show the uncertainty in the measurement. Presumabley repeat measurements were made because there are standard deviations (standard error?) provided in Table 1.

**Response:**

Thanks for this suggestion. We have done accordingly.

Table 1: Please include all the quantities measured and calculated for each sample (e.g. AAE, $R_{BrC/BC}$, $f_{BrC}$, $F_{BrC}$) along with propagated uncertainties.

**Response:**

Thanks for this suggestion. The AAE data can be found in Table S2-I, and the $R_{BrC/BC}$ data were added in Table 1. The data of $f_{BrC}$ were both sample-specific (11 biomass fuels and more than 20 coals) and wavelength-specific (stepwise from 350 nm to 850 nm) and therefore were too many; we had to arrange them (for biomass fuels and coals (Sun et al., 2017)) in Table S2-II (Supporting Information). The plots of $f_{BrC}$ for biomass fuel and coal can be seen in Figure 4.

$F_{BrC}$ is just sample-specific and can't be given in every single wavelength like $f_{BrC}$.

Figure 4: What is the impact of limit of detection on this plot? The data > 750 nm is very noisy, and I wonder if that is not due to limitations of the measurement? If this data is below the limit of detection it should not be used in the calculation of $f_{BrC}$.

**Response:**

Thanks for this reminder. The samples of coals and biomass fuels were actually analyzed with the same instrument (Perkin Elmer Lambda 950) during the same period. The status of the instrument was normal and stable then. In Figure 4, although the $f_{BrC}$ at wavelength >750 nm for biomass fuels looked very "noisy", the $f_{BrC}$ at wavelength >750 nm for coals fluctuated very gently. This implies that the larger fluctuation for biomass fuels than for coals in Figure 4 resulted unlikely from the limitation of measurement (instrumental detection limit) but very likely from samples themselves (e.g, chemical composition). Sure it deserves further study in future. Again thanks.

Figure 5 and line 303: Why only use the mean (median?) $f_{BrC}$ from these current study and Sun et al. 2017? Although the regression is strongly correlated here, the scatter in the data is covered up by using the mean value instead of every measured data point.

**Response:**

The question why only the mean values of $F_{BrC}$ vs AAE rather than the data of each sample were used to construct the function between $F_{BrC}$ and AAE is really worth explaining. The same question had actually been raised by the editor and we had listed the reasons. On the one hand, we know, each of the latter three points (1.58, 0.265; 2.46, 0.508; 6.09, 1.00) in Figure 5 is the average of a number of data, and therefore each of them can be replaced with a cluster of individual dots if we like; yet on the other hand, the first point (0.00, 1.00) is not originated from averaging over a cluster of individuals but from theoretical consideration, and hence there are no cluster of individual dots to replace this single point. Under the circumstances, replacing each of the latter three points with a cluster of individual dots will substantially lower the weight of the first point from 25% to almost being negligible. Given the theoretical significance of the first point, this is not only unfair but also unacceptable. For this consideration, we prefer to use the average value for each of the latter three points so that all the four points in Figure 5 are put equal weight (25%). Additionally, compared with a cluster of individuals, an average is usually more representative of the true value and hence is more persuasive. We added an explanation in our revised version (lines 340-341).

Uncertainty exists in every pairs and thus in the algorithm. For example, for pure BC (the first pair), Lack and Langridge (2013) estimated that the uncertainty in short wavelength absorption by

BC determined by extrapolation using an AAE=1 ranged from +7% to −22%. The other 3 points in

Figure 5 are all averages over a cluster of individual dots and therefore we are able to give error bars for every points (Figure 5).

**SI:**

Tables S3 Part I: Extracts are dominated by ambient aerosols, what about source? E.g. fossil fuel combustion, woodstoves, open burning? (Just a few examples are: Xie et al. 2017

https://doi.org/10.1038/s41598-017-06981-8 for open burning and gasoline exhaust; Xie et al. 2018

https://doi.org/10.1016/j.envpol.2018.04.085 for cookstoves using wood, kerosene and charcoal;

Corbin et al. 2018 https://doi.org/10.1029/2017JD027818 for marine diesel engines). Since this paper is focused on emissions it would be good to have a more exhaust list of emissions AAE

measurements. Calculations: should 'coal' here be 'biomass fuel'?

**Response:**

Thanks for this reminder. We agree that "Since this paper is focused on emissions it would be good to have a more exhaust list of emissions AAE measurements". In our revised version, the suggested AAEs have been added to Table S3 part I.

We are grateful for the reviewer's carefulness in finding our miswording and have changed

'coal' to 'biomass fuel'.

Figures S2-S4: Appear to be identical to those in Sun et al. 2017, should the reference be noted in the caption?

**Response:**

Thanks. We have added the reference of 'Sun et al., 2017' in Figure S2-S4.

Figure S4: Hard to follow the text here, would be easier to understand in equation form or even a diagram.

**Response:**

Thanks for this suggestion. We'd like to add a flow chart in Supporting Information (Figure S5), so that readers could understand the mechanism of iterative process used in this study more easily.

[Figure]

Figure S5 Calculation of BC and BrC with iterative process

**2.4 Calculation methods**

[revised manuscript text omitted]

Figure 1 aids to compare $EF_{BrC}$ and $EF_{BC}$. The ratios of $EF_{BrC}$ to $EF_{BC}$ ($R_{BrC/BC}$) varied greatly among various biomass fuels and corncobs and sorghum stalks gave the highest (10.0) and lowest (1.5) $R_{BrC/BC}$

values, respectively. Generally, the large rang of $R_{BrC/BC}$ values among different biomass fuels is attributable to the individual biomass fuels themselves, or more concretely their chemical composition and physical structure. Here both BrC and BC were products of incomplete combustion of biomass fuels (Andreae and Gelencsér, 2006. Yan et al., 2015). Different biomass fuels were composed of different organics that had different combustion performances (Reid et al., 2005; Saleh et al., 2014); meanwhile, different biomass fuels were also different in densities and moistures (Shen et al., 2014;

Jacobson et al., 2015), which also have a potential influence on combustion performance. The combustion performance relates to something like the combustion speed and temperature, both of which are important to the formation of BrC and BC. Usually a low combustion temperature is more favorable for BrC formation and a relatively high combustion temperature is more favorable for BC

formation (Chen and Bond, 2010; Bond et al., 2013; Shen et al., 2014). This makes the generation processes of BC and BrC often not synchronous but in opposite trend, which may account for wide variations of $R_{BrC/BC}$ for different fuels of combustion conditions.

More importantly, each of the 11 biomass fuels tested in this study had a higher $EF_{BrC}$ than $EF_{BC}$; that is, the ratios of $EF_{BrC}$ to $EF_{BC}$ ($R_{BrC/BC}$) were all >1. The average $R_{BrC/BC}$ over all biomass fuels was

$6.7 \pm 2.7$. Kirchstetter et al. (2004) measured the light absorption by filter-based aerosol samples from biomass burning before and after acetone treatment (which removed OC). They found that 50% of total light absorption was attributable to OC. In view of the much smaller average absorption efficiency of

BrC relative to that of BC (for example, Yang et al. (2009) reported that the MAEs at 550 nm were 9.5,

0.5, and 0.03 $m^2$/g, respectively, for BC, BrC, and dust), the contribution of BrC to the mass of total

[revised manuscript text omitted]

2016; Lei et al., 2018b) (Table S3 Part I). The other two pairs of the $F_{BrC}$ vs AAE values were obtained from our previous and current studies. The previous study (Sun et al., 2017) demonstrated that, when

AAE was 1.58, $F_{BrC}$ was 0.265. In the present study, as mentioned in Section 3.3, an AAE of 2.46 led to an $F_{BrC}$ of 0.508. These four $F_{BrC}$ vs AAE pairs were used to construct the relationship between $F_{BrC}$

and AAE (Figure 5). It should be noted that we used the average value for each of the latter three points so that all the four points in Figure 5 were given equal weight (25%). A logarithmical equation was established between $F_{BrC}$ and AAE, with a very high correlation coefficient.

$F_{BrC} = 0.5519 \ln AAE + 0.0067$         ($R^2 = 0.999$)                 (2)

Equation (2) provides a novel algorithm for deriving $F_{BrC}$ from AAE, without consideration of the process details for any kinds of combustion sources. Uncertainties are unavoidable due to the uncertainties of each of the points (Lack and Langridge, 2013; Sun et al., 2017; references in Part I of

Table S3). For example, Lack and Langridge (2013) estimated that the uncertainty in short wavelength absorption by BC determined by extrapolation using an AAE=1, ranged from +7% to −22%. Equation (2) helps to broaden insight into biomass burning issues from contained conditions to open conditions.

The results of $F_{BrC}$ for open fresh emissions from open biomass burning ($F_{BrC-open}$) vary in the literature, and most have values below 0.50 (or 50%) (Lack et al., 2012; Healy et al., 2015; Washenfelder et al.,

2015; Srinivas, et al., 2016). We collected AAE$_{-open}$ data from available journal articles and included them in Table S3 (Part II). The calculated average AAE$_{-open}$ value was 3.44 ± 1.75, which was larger than the AAE$_{-contained}$ value obtained in this study (2.46 ± 0.53). Substitution of the AAE$_{-open}$ value (3.44

± 1.75) into Equation (2) leads to a value of 0.685 for $F_{BrC-open}$, which is higher than the $F_{BrC}$ for contained combustion ($F_{BrC\ -contained}$) (0.508), indicating that BrC's light absorption was more dominant in open biomass burning emissions than in contained biomass burning emissions.

Assuming that the AAE$_{-contained}$ and AAE$_{-open}$ identified above apply to global biomass burning, we can now assess BrC's role in the biomass burning globally (contained + open) ($F_{BrC-entire}$), in combination with the respective shares of open and contained burning. Previous studies show that the annual open and contained biomass burning amounts are 5953 Tg (Wiedinmyer et al., 2011) and 2457

Tg (Fernandes et al., 2007), respectively. This implies that open biomass burning represents 71% of total biomass burning and contained biomass burning represents 29%. Subsequently, the $F_{BrC-entire}$ can be calculated according to the following equation:

$$F_{BrC-entire} = 0.29 \times (0.5519 \ln AAE_{-contained} + 0.0067) + 0.71 \times (0.5519 \ln AAE_{-open} + 0.0067) \qquad (3)$$

With Equation (2), the distribution of $F_{BrC-entire}$ was simulated through the Monte Carlo approach, as shown in Figure 6. The $F_{BrC-entire}$ was 0.644 on average, and with an 80% probability range it lay between 0.585–0.699. Particularly, the probability of $F_{BrC-entire}$ being larger than 0.500 was higher than

99%, corroborating the leading role of BrC in the absorption by solar light for total biomass burning emissions. Kirchstetter and Thatcher (2012), calculate that OC from wood smoke would account for

14% of solar radiation absorbed by wood smoke in the atmosphere (integrated over the solar spectrum from 300 to 2500 nm). 14% is much smaller than out data $F_{BrC-entire}$= 64.4% because Kirchstetter and

Thatcher (2012) only focus on rural California wintertime wood combustion but we calculated the global contribution to absorption by BrC originating from biomass combustion.

**4 Conclusions**

The optical IS approach was used to distinguish BrC from BC in filter samples of the emissions of

11 types of biomass after burning in a typical stove. The measured average EF of household biomass fuels for BrC was 0.71 g/kg, and the calculated annual BrC emissions from China's household biomass burning amounted to 712 Gg. This is higher than the emissions from China's household coal combustion (592 Gg). Moreover, it was observed that BrC contributed to approximately half of all light absorption by BC + BrC across the strongest solar spectral range (350–850 nm; $F_{BrC}$ = 50.8%).

Furthermore, a novel relationship was constructed ($F_{BrC}$ = 0.5519ln(AAE) + 0.0067, $R^2$ = 0.999), which can simplify the calculation of $F_{BrC}$ by using AAE. With this mathematical relationship, we calculated the $F_{BrC}$ values for open biomass burning ($F_{BrC-open}$ = 70.1%) and entire biomass burning ($F_{BrC-entire}$ =

64.4%), thereby establishing the dominant role of BrC in biomass burning absorption. From this perspective, we recommend that it is necessary to include BrC in the climate discussion, particularly concerning biomass burning (contained and open). The algorithm developed here omits the long procedures of chemical treatment, optical measurement and tedious calculations, and provides a scheme for estimating the contribution of BrC relative to BC in perhaps any combustion process with LAC emissions.

**Data availability**

The research data can be accessed, on request, from the corresponding author (zhigr@craes.org.cn).

**Acknowledgements**

This study was supported by the National Natural Science Foundation (41977309), Research results of 13th five-year plan for Social Sciences in Jiangxi Province, China (19ZK34), National Key Research & Development Plan (2017YFC0213001), Special Project of Fundamental Research Funds of the Chinese Research Academy of Environmental Sciences (JY41373131), Chinese-Norwegian Project on Emission, Impact, and Control Policy for Black Carbon and its Co-benefits in Northern China (CHN-2148-19/0029), and Topics of Jiangxi Sports Bureau, China (2018021).

*Competing interests.* The authors declare that they have no conflicts of interest.

[revised manuscript text omitted]

2.46 ± 0.16 (mean ± SD of the means) and AAE$_{\text{-open}}$ value of 3.44 ± 0.42 (mean ± SD of the means)

apply to whole world biomass burning, the combined value for entire biomass burning ($F_{\text{BrC-entire}}$) can be calculated as: $F_{\text{BrC-entire}} = 0.71 \times (0.5519 \ln \text{AAE}_{\text{-open}} + 0.0067) + 0.29 \times (0.5519 \ln \text{AAE}_{\text{-contained}} +$

0.0067)

---

## Referee Report (RR1)

Additional minor comments:

- Line 121 – Instead of "envisaged emissions intensity of each combination process" state "desired PM2.5 concentration in the dilution system" that way the reader can understand what criteria was used and present the target concentration as well. The sampling concentration is an important factor in the partitioning of semivolatile species, which may be contributing to BrC absorption that was collected on the filter. There is semi-volatile BrC – see Xie et al. 2020 https://doi.org/10.5194/acp-20-14077-2020 - which may end up being collected on the filter depending upon the sampling concentrations.

- Line 168 – 171 – The authors note that their reference material CarB has an AAE of 0.91 and HASS has an AAE of 1.86. It is problematic that the HASS reference material has an AAE that is far lower than the BrC values listed in table S3. The authors need to discuss the implication of using a reference material with a much lower AAE than other BrC sources. The authors should also report the MAE for each of the reference materials to facilitate comparisons with other approaches to quantify BrC mass.

- Lines 200 – 220 – The authors should present the BrC EFs from Shen et al. 2013, that was referenced on line 194-197.

- Line 289 – The authors should include a reference for the source of the funeral pyre emissions estimate.

- Figure 4 – How can you state that the absorption of the samples was above the limit of detection at wavelengths in the 750 – 850 nm range when absorption for the reference material was only quantified at 650 nm? For combustion generated PM the absorption generally decreases as the wavelength increases so being above the limit of detection at 650 nm does not imply that the measurement is above the limit of detection in the range of 750 – 850 nm. Also, there should be error bars on this figure since this is representing multiple samples.

- Equation 2 and Figure 5 – It should be noted that this equation only applies to a range of AAEs, since AAE's larger than their max value will result in fractions greater than 1.

---

## Author Response (AR2)

**Reply to Editor,**

**EDITORIAL REVIEW:**

1. L343: the 4 points in figure 5 fall on a straight line. I am curious why you elect to complicate the fit by using a logarithmic function? If you insist on using logarithmic fit you need to provide a theoretical justification for it. If there is no theoretical reason, and the fit is purely empirical, using the simplest possible function is better.

**Response:**

Thanks for this comment. The fit in Figure 5 sure was from the data of our studies and existing literature and is basically empirical. Although the 4 points in the Figure looks like a straight line, it actually is a curve as the x axis is not scaled uniformly but geometrically.

As for why we expressed the empirical relation with a logarithmic function instead of others is (i) when AAE=1 (pure BC),  $F_{BrC} = 0$ , which complies with the property of logarithmic function, i.e., log1.0 = 0 and (ii) the fitted logarithmic function displayed a significant correlativity between  $F_{BrC}$  and AAE ( $F_{BrC} = 0.5519$ lnAAE + 0.0067, R2 = 0.9998).

Given the above facts, in our newest version, we changed geometrically scaled x axis to uniformly scaled x axis while maintained the logarithmic relation in Figure 5, as below:

2. Table 1: Some digits can be cut from numbers without loss of information. For example,  $2.50 \pm 3.064$  can be replaced by  $2.50 \pm 3.06$  as there is no reason to specify error in the digit that is not even listed. Same for  $1.25 \pm -0.074$ ,  $1.51 \pm -0.389$ , etc. As I mentioned in the initial review, your EF error for rape straw is unphysically small leading to unnecessarily precise EFs reported here. I am pretty sure that using more conservative measurements uncertainties would increase the reported error, perhaps from 0.002 to more than 0.01 (allowing you to cut one significant digit out). Are you comfortable reporting an emission factor with 4 significant digits when the rest of them have 2-3?

**Response:**

Thanks for this suggestion. We followed the suggestion and replaced  $2.50 \pm 3.064$  with  $2.50 \pm 3.064$ , replaced  $1.25 \pm 0.074$  with  $1.25 \pm 0.07$ , and replaced  $1.51 \pm 0.389$  with  $1.51 \pm 0.39$ . Actually we examined all the data in Table 1 and cut the digits after the decimal point to two. This includes the data for the rape straw from  $7.259 \pm 0.002$  to  $7.26 \pm 0.01$  (EFBrC), from  $2.537 \pm 0.001$  to  $2.54 \pm 0.01$  (EFBC), and from  $2.86 \pm 0.018$  to  $2.86 \pm 0.02$  (RBrC/BC).

3. L118: Use of alcohol presumably makes MCE higher than it would have been without it. Should it be mentioned?

**Response:**

We agree to this comment and mentioned it in our newest version (lines 121-123).

4. L206: This is an important practical recommendation from your study, and if this is not widely known it may be worth inserting a sentence about the benefits of biomass briquetting in the abstract.

**Response:**

Thanks for this suggestion. In the Abstract we have inserted a sentence about the benefits of biomass briquetting (lines 12-14).

TECHNICAL CORRECTIONS:

5. L11: geomean -> geometric mean (also make this change in other places in the text such as Table 1)

**Response:**

Thanks for this reminder. We checked throughout the manuscript and changed "geomean" to "geometric mean".

6. L92: Eleven biomass fuels were tested: they were classified into three groups, i.e. crop -> Eleven biomass fuels tested in this work were classified into three groups: crop

**Response:**

Thanks for this suggestion. We have done accordingly in line 94-95.

7. L111: 93.86 ± 5.93% -> 93.9 ± 5.9%,

**Response:**

Thanks for the suggestion. All similar cases have been updated accordingly, as follows:

- 83.95% (line 110 in previous version) was changed to 84.0% (line 112 in newest version)
- $93.86 \pm 5.93\%$  (line 111 in previous version) to  $93.9 \pm 5.9\%$  (line 113 in newest version)

8. L141: (e.g. Acros -> (Acros

**Response:**

Thanks for this suggestion. We have done accordingly (newest version, line 147)

9. L154: not fully perfect -> not perfect

**Response:**

Thanks for this suggestion. We have deleted 'fully' in the sentence (newest version, line 160).

**10. L221: aids to compare -> compares**

**Response:**

We have changed 'aids to compare' to 'compares' (newest version, line 224).

11. L274: are collated and arranged in a scatter plot (Figure 2) -> are collated and arranged in a scatter

plot in Figure 2

**Response:**

Thanks for this suggestion. We have done accordingly (newest version, line 277).

**12. L327: logarithmical -> linear**

**Response:**

Thanks for this suggestion. We actually kept the logarithmical relation unchanged (newest version, line 344). In response to editor's comment 1, we have explained why we maintained the logarithmic relation for Figure 5.

Again we thank the editor for above suggestions, which are very helpful to further improve our manuscript.

**Reply to Reviewer 1,**

**Additional minor comments:**

Many thanks for recommending publication of our manuscript after the following minor comments are addressed. We have accordingly revised our manuscript, as follows.

1. Line 121 - Instead of "envisaged emissions intensity of each combination process" state "desired PM2.5 concentration in the dilution system" that way the reader can understand what criteria was used and present the target concentration as well. The sampling concentration is an important factor in the partitioning of semi-volatile species, which may be contributing to BrC absorption that was collected the filter. There is semi-volatile BrC Xie et al. 2020 on see https://doi.org/10.5194/acp-20-14077-2020 -which may end up being collected on the filter depending upon the sampling concentrations.

**Response:**

Thanks for this insightful comment. It does be important to arrange an appropriate dilution factor for each biomass fuel so that the light absorbance of absorbing aerosols (BrC+BC) deposited on a filter falls in the linear range of integrating sphere (IS) approach. Pre-experiments were carried out in advance to decide on the weight of a biomass fuel to be burned and the ratio of a dilution to be set. For each biomass fuel, the weight burned and the ratio diluted are both presented in Table S1-II in the newest version of Supplement. We also gave a brief description of the principle and specific arrangement of flue gas diluting in lines 125-126 (newest version).

We agree to the effect of the sampling concentration on the partitioning of semi-volatile species and added a new sentence in lines 127-128 (newest version) to highlight the implications.

2. Line 168 – 171 – The authors note that their reference material CarB has an AAE of 0.91 and HASS has an AAE of 1.86. It is problematic that the HASS reference material has an AAE that is far lower than the BrC values listed in table S3. The authors need to discuss the implication of using a reference material with a much lower AAE than other BrC sources. The authors should also report the MAE for each of the reference materials to facilitate comparisons with other approaches to quantify BrC mass.

**Response:**

Thanks for this meaningful comment. We also have noticed that the HASS reference material used in our study had an AAE that was far lower than the BrC values listed in table S3.

Since BrC is not a pure substance but a collection of light-absorbing organic substances, the AAE values of BrC of individual samples vary a lot depending on what light absorbing organic substances comprise the BrC. In this sense, it's actually impossible to identify a substance that can precisely represent all BrC substances. However, the need for quantification of BrC implies a need for such a material that is close in some aspects to ordinary BrC species and can be used as a reference or standard in some degree. To this end, the past decades saw HASS being chosen as the reference of BrC (e.g., Wonaschütz et al., 2009; Sun et al., 2017) just as the carbon black (CarB) was chosen as the reference of BC (e.g., Medalia et al., 1983; Hitzenberger et al., 1996; Hitzenberger et al., 2001, 2006; Reisinger et al., 2008 ). HASS is an organic substance that has some similar properties to BrC (e.g., brown color, with an AAE value apparently higher than 1.0). Moreover HASS is chemically stable and water-soluble, and can be conveniently used for the preparation of a standard solution for calibration purpose.

The AAE of HASS is more than twice that of CarB, which suffices the iterative calculation between two designated wavelengths: 365 nm and 650 nm. In this study (newest version, lines 153-162), we stated that 'In the present study, we continued this logic, and assumed that BC and BrC in household biomass smoke have the same light-absorbing properties as CarB and HASS, respectively. In other words, the reported BC and BrC masses here are essentially CarB-C-equivalent and HASS-C-equivalent, respectively, from the perspective of light absorption and are different from those measured by other measurement techniques (e.g., thermal–optical method or aethalometer) (Chen et al., 2006; Zhi et al., 2008, 2009; Shen et al., 2013, 2014; Aurell and Gullett, 2013) or reference materials (e.g., fulvic acid, humic acid, or humic-like substances) (Duarte et al., 2007; Lukács, et al., 2007; Baduel et al., 2009, 2010). Although such an assumption is not perfect, researchers can take advantage of these two reference materials to relatively quantify and assess the features (chemical or optical) of BrC and BC derived from different combustion sources or regions'.

The MAE values of HASS and CarB were given in lines 174-177, respectively (newest version).

3. Lines 200 – 220 – The authors should present the BrC EFs from Shen et al. 2013, that was referenced on line 194-197.

**Response:**

Thanks for this suggestion. We checked the paper published by Shen et al. (2013) and found no BrC EFs explicitly or implied. Very sorry for the mis-citing. We have deleted the text 'A similar phenomenon was also observed by Shen et al. (2013), who carried out a systematic measurement of PM, OC, and EC released from various solid fuels burned in residential stoves; these authors found that crop residues, which were composed of herbaceous plants, were more likely to have higher BrC EFs than wood fuels, which were composed of ligneous plants' (lines 193-197 in previous version).

4. Line 289 – The authors should include a reference for the source of the funeral pyre emissions estimate.

**Response:**

Thanks for the suggestion. We added a reference for the source of the funeral pyre emissions estimate (newest version, line 293).

5. Figure 4 – How can you state that the absorption of the samples was above the limit of detection at wavelengths in the 750 - 850 nm range when absorption for the reference material was only quantified at 650 nm? For combustion generated PM the absorption generally decreases as the wavelength increases so being above the limit of detection at 650 nm does not imply that the measurement is above the limit of detection in the range of 750 - 850 nm. Also, there should be error bars on this figure since this is representing multiple samples.

**Response:**

Thanks a lot for this comment. We acknowledge that being above the limit of detection at 650 nm does not necessarily imply that the measurements of combustion samples are above the limit of detection in the range of 750 – 850 nm because the absorption of combustion generated PM generally decreases as the wavelength increases. This didn't influence the results of emission factors and the overall profiles of wavelength dependent  $f_{BrC}$  (The emission factors obtained through iterative calculation are only relevant to the wavelengths of 365 nm and 650 nm).

In the newest version, we added error bars not only on the blue line ( $f_{BrC}$  for biomass in this study) but also on the red line ( $f_{BrC}$  for household coal in previous study [Sun et al., 2017]) in Figure 4.

6. Equation 2 and Figure 5 – It should be noted that this equation only applies to a range of AAEs, since AAE's larger than their max value will result in fractions greater than 1.

**Response:**

Thanks for this suggestion. The paragraph of lines 333-345 describes how Equation 2 is derived, which implies the domain of the function ranges from AAE = 1.0 (first point) to AAE = 6.09 (last point). We gave explanatory note after the expression of Equation 2 (see newest version, line 346).

**References:**

- Aurell, J. and Gullett, B. K.: Emission factors from aerial and ground measurements of field and laboratory forest burns in the southeastern US: PM2.5, black and brown carbon, VOC, and PCDD/PCDF, Environ. Sci. Technol., 47(15), 8443-8452, doi: 10.1021/es402101k, 2013.
- Baduel, C., Voisin, D., and Jaffrezo, J. L.: Comparison of analytical methods for Humic Like Substances (HULIS)measurements in atmospheric particles, Atmos. Chem. Phys., 9, 5949–5962, doi: 10.5194/acp-9-5949-2009, 2009.
- Baduel, C., Voisin, D., and Jaffrezo, J. L.: Seasonal variations of concentrations and optical properties of water solubleHULIS collected in urban environments, Atmos. Chem. Phys., 10, 4085–4095, doi: 10.5194/acp-10-4085-2010, 2010.
- Chen, Y., Zhi, G., Feng, Y., Fu, J., Feng, J., Sheng, G., and Simoneit, B. R. T.: Measurements of emission factorsfor primary carbonaceous particles from residential raw-coal combustion in China, Geophys. Res. Lett., 33 (20), 1-4, doi: 10.1029/2006gl026966, 2006.
- Duarte, R. M., Santos, E. B., Pio, C. A., and Duarte, A. C.: Comparison of structural features of water-soluble organic matter from atmospheric aerosols with those of aquatic humic substances, Atmos. Environ., 41(37), 8100–8113, doi: 10.1016/j.atmosenv.2007.06.034, 2007.
- Hitzenberger, R., Dusek, U., and Berner, A.: Black carbon measurements using an integrating sphere, J. Geophys. Res., 101(D14), 19601-19606, doi: 10.1029/95jd02412, 1996.
- Hitzenberger, R. and Tohno, S., Comparison of black carbon (BC) aerosols in two urban areas-concentrations and size distributions, Atmos. Environ., 35, 2153-2167, doi: 10.1016/s1352-2310(00)00480-5, 2001.
- Hitzenberger, R., Peezold, A., Bauer, H., Ctyroky, P., Pouresmaeil, P., Laskus, L., and Puxbaum, H.: Intercomparison of thermal and optical measurement methods for elemental carbon and black carbon at an urban location, Environ. Sci. Technol., 40, 6377-6383, doi: 10.1021/ed051228v, 2006.
- Lukács, H., Gelencsér, A., Hammer, S., Puxbaum, H., Pio, C., Legrand, M., Kasper-Giebl, A., Handler, M., Limbeck, A., Simpson, D., and Preunkert, S.: Seasonal trends and possible sources of brown carbon based on 2-year aerosol measurements at six sites in Europe, J. Geophys. Res., 112 (D23S18), 1-9, doi: 10.1029/2006JD008151, 2007.
- Medalia, A. I., Rivin, D., and Sanders, D. R.: A comparison of carbon black with soot, Sci. Total Environ., 31, 1-22, doi: 10.1016/0048-9697(83)90053-0, 1983.
- Reisinger, P., Wonaschütz, A., Hitzenberger, R., Petzold, A., Bauer, H., and Jankowski, N.: Intercomparison of measurement techniques for black or elemental carbon under urban background conditions in wintertime: Influence of biomass combustion, Environ. Sci. Technol., 42, 884-889, doi: 10.1021/es0715041, 2008.
- Shen, G., Tao, S., Wei, S., Chen, Y., Zhang, Y., Shen, H., Huang, Y., Zhu, D., Yuan, C., Wang, H., Wang, Y., Pei, L., Liao, Y., Duan, Y., Wang, B., Wang, R., Lv, Y., Li, W., Wang, X., and Zheng, X.: Field measurement of emission factors of PM, EC, OC, parent, nitro-, and oxy- polycyclic aromatic hydrocarbons for residential briquette, coal cake, and wood in rural Shanxi, China, Environ. Sci. Technol., 47 (6), 2998-3005, doi: 10.1021/es304599g, 2013.
- Shen, G., Xue, M., Chen, Y., Yang, C., Li, W., Shen, H., Huang, Y., Zhang, Y., Chen, H., Zhu, Y., Wu, H., Ding, A. and Tao, S.: Comparison of carbonaceous particulate matter emission factors among different solid fuels burned in residential stoves, Atmo. Environ., 89, 337-345, doi: 10.1016/j.atmosenv.2014.01.033, 2014.

Sun, J., Zhi, G., Hitzenberger, R., Chen, Y., Tian, C., Zhang, Y., Feng, Y., Cheng, M., Zhang, Y., Cai, J.,

Chen, F., Qiu, Y., Jiang, Z., Li, J., Zhang, G., and Mo, Y.: Emission factors and light absorption properties of brown carbon from household coal combustion in China, Atmos. Chem. Phys., 17(7), 4769-4780, doi: 10.5194/acp-17-4769-2017, 2017.

- Wonaschütz, A., Hitzenberger, R., Bauer, H., Pouresmaeil, P., Klatzer, B., Caseiro, A., and Puxbaum, H.: Application of the integrating sphere method to separate the contributions of brown and black carbon in atmospheric aerosols, Environ. Sci. Technol., 43, 1141-1146, doi: 10.1021/es8008503, 2009.
- Zhi, G., Chen, Y., Feng, Y., Xiong, S., Li, J., Zhang, G., Sheng, G., and Fu, J.: Emission characteristics of carbonaceous particles from various residential coal-stoves in China, Environ. Sci. Technol., 42(9), 3310-3315, doi: 10.1021/es702247q, 2008.
- Zhi, G., Peng, C., Chen, Y., Liu, D., Sheng, G., and Fu, J.: Deployment of coal briquettes and improved stoves: possibly an option for both environment and climate, Environ. Sci. Technol., 43(15), 5586-5591, doi: 10.1021/es802955d, 2009.